# Activation of the ciliary kinase CDKL5 is mediated by the cyclin-dependent kinase CDK20/LF2 to control flagellar length

Yuqing Hou[1☯], Oranti Ahmed Omi[2☯], Michael W. Stuck[3], Xi Cheng[1], Bethany Walker[1], Ying-Wai Lam[4], Anna M. Schmoker[5¤], Son N. Nguyen[6], Maria Paz Gonzalez-Perez[6], Bryan A. Ballif[4], Karl F. Lechtreck[2]*, George B. Witman[1]*, Gregory J. Pazour[3]*

1 Department of Radiology, UMass Chan Medical School, Worcester, Massachusetts, United States of America, 2 Department of Cellular Biology, University of Georgia, Athens, Georgia, United States of America, 3 Program in Molecular Medicine, UMass Chan Medical School, Worcester, Massachusetts, United States of America, 4 Department of Biology, University of Vermont, Burlington, Vermont, United States of America, 5 Dana-Farber Cancer Institute, Boston, Massachusetts, United States of America, 6 Mass Spectrometry Facility, UMass Chan Medical School, Shrewsbury, Massachusetts, United States of America

☯ These authors contributed equally to this work.
¤ Current address: Biological Mass Spectrometry and Proteomics Shared Resource, Geisel School of Medicine, Dartmouth College, Hanover, New Hampshire, United States of America
* george.witman@umassmed.edu (GBW); lechtrek@uga.edu (KFL); gregory.pazour@umassmed.edu (GJP)

## Abstract

Variants in the protein kinase CDKL5 cause CDKL5 Deficiency Disorder (CDD), a severe neurodevelopmental condition characterized by seizures, developmental delay, and intellectual disability. The *Chlamydomonas* homolog of CDKL5, LF5, is a flagellar protein whose loss leads to elongated flagella. Here, we combine live-cell imaging, immunofluorescence, and biochemical approaches including mass spectrometry to define how CDKL5 activity is regulated and how its loss alters ciliary function. We find that *Chlamydomonas* CDKL5 is activated by LF2, a cyclin-dependent kinase, through phosphorylation of its activation loop. This activation controls CDKL5 localization in steady-state cilia, down-regulates its IFT-mediated transport as flagella reach steady-state, controls ciliary abundance of IFT proteins, and controls phosphorylation of the tubulin-binding domain of IFT74, thereby influencing flagellar length. Mouse Cdkl5 shows similar properties: it localizes within cilia, its loss leads to ciliary elongation, and its localization depends on both its kinase activity and Cdk20, the mammalian ortholog of LF2. These results extend our understanding of ciliary length control, challenge the prevailing model that CDKL5 is activated by autophosphorylation, and suggest that CDD pathogenesis arises, at least in part, from disruption of this conserved ciliary regulatory pathway.

**Data availability statement:** MS datasets are available from the Pride database (https://www.ebi.ac.uk/pride/) under accession numbers PXD066796, PXD066877, and PXD068782.

**Funding:** This work was supported by grants from the Loulou Foundation (GJP), the Li Weibo Institute for Rare Diseases Research at the University of Massachusetts Chan Medical School (GJP), and the National Institutes of Health (R35GM122574 to G.B.W., GM060992 to G.J.P., and R35GM152057 to K.F.L.). In addition, the UMass Chan Department of Radiology provided support (Y.H. and G.B.W.). The funders had no role in study design, data collection and analysis, decision to publish, or preparation of the manuscript.

**Competing interests:** The authors have declared that no competing interests exist.

**Abbreviations:** aa, amino acids; ATPγS, adenosine-5′-O-(3-thiotriphosphate); Bsd, blasticidin; CB, cell body; CDD, CDKL5 Deficiency Disorder; CDKL, cyclin-dependent kinase-like; CIP, calf-intestinal phosphatase; IFT, intraflagellar transport; MS, mass spectrometry; MEF, mouse embryonic fibroblast; Nat, nourseothricin; Paro, paromomycin; Puro, puromycin; TAP, Tris-acetate-phosphate; TIRF, total internal reflection fluorescence; TMT, tandem mass tag; Hyg, hygromycin.

## Introduction

CDKL5 deficiency disorder (CDD) causes seizures, developmental delay, and severe intellectual disability in affected individuals. In addition, patients with CDD variably present with cranial facial and hand anomalies, have significant gastrointestinal dysfunction and sleep disorders along with recurrent pneumonia and respiratory disease [1]. The affected gene is X-linked, and most patients are females. CDKL5 is a protein kinase that is conserved across ciliated organisms including the green alga *Chlamydomonas reinhardtii*, where CDKL5 localizes to flagella and its loss causes the flagella to be abnormally long [2].

Cilia and flagella are evolutionarily conserved, microtubule-based organelles that serve diverse sensory and motility functions throughout the eukaryotic kingdom. We will use the terms "flagella" when referring to *Chlamydomonas* and "cilia" when referring to mammals or making a generalization. In mammals, dysfunction of primary cilia causes complex developmental and degenerative phenotypes affecting multiple organs. Organ involvement varies depending on the gene affected and the strength of the allele. Phenotypes common in CDD, including intellectual disability, brain abnormalities, and skeletal dysplasias such as hand and facial dysmorphias, are often observed in patients with primary cilia defects [3–5]. In addition, CDD patients have significant morbidity due to respiratory infections, a phenotype shared with patients with motile cilia dysfunction [1,6]. Finally, defects in both motile and primary cilia have been connected to epilepsy [7,8]. This phenotypic overlap together with the knowledge that CDKL5 is a ciliary protein in both *Chlamydomonas* and mammals [9–11] suggests that CDD is a ciliopathy and that understanding the function of CDKL5 in cilia will be key to understanding the pathology of this devastating disease.

The first indication that CDKL5 was a ciliary protein came from proteomic analysis of *Chlamydomonas* flagella, where the homolog, then referred to as FAP247, was enriched in a KCl-extract of the flagellum's microtubular cytoskeleton, or axoneme, and its transcript was upregulated during flagellar growth, a property of many ciliary genes [12]. These findings were extended by Tam and colleagues [2], who discovered that mutants null for *FAP247* had flagella about 1.5× normal length, indicating that the gene product has a role in flagellar length control. Four other *Chlamydomonas* genes (*LF1*, *LF2*, *LF3*, and *LF4*) involved in flagellar length control were already known, so Tam and colleagues renamed the gene *LF5*. We will here refer to the FAP247/LF5 protein as CrCDKL5 or simply CDKL5. Tam *and colleagues* showed that CrCDKL5 is distributed along the length of the growing flagellum and then becomes concentrated at the proximal end just distal to the transition zone as the flagellum approaches steady-state where growth has leveled off. In mammals, evidence that CDKL5 is connected to cilia include the findings that human CDKL5 is localized to cilia in cultured cells [9], that the centriolar satellite protein CEP131 [13,14] is a substrate of CDKL5 [15,16], that *Cdkl5* mutant mice have elongated cilia in the hippocampus [10], and that a *Cdkl5* mutation increases cilia length in respiratory and ependymal cells and alters cerebral spinal fluid flow [11].

CDKL5 is a member of the cyclin-dependent kinase like (CDKL) subfamily of a larger family that includes mitogen-activated protein kinases, glycogen synthase

kinases, and CDK kinases. These enzymes contain a kinase catalytic domain consisting of an ATP-binding site, an active site lysine, and a TEY activation loop. The TEY activation loop in mammalian CDKL5 is thought to be auto-phosphorylated to control activity [16–19].

In this work, we show that *Chlamydomonas* CDKL5 is highly phosphorylated across its length, including residues within the activation loop. Unlike the prevailing model in mammals, *Chlamydomonas* CDKL5 kinase activity is not required for activation loop phosphorylation. Instead, this modification depends on LF2, a cyclin-dependent kinase family member [20]. Consistent with this finding, loss of LF2 phenocopies the effects of CDKL5 kinase inactivation. We further demonstrate that CDKL5 is transported by intraflagellar transport (IFT) during flagellar growth, and that this transport is down-regulated once flagella reach steady-state. This regulation requires both LF2 and CDKL5 kinase activity, as elevated transport is maintained when either is disrupted. Our data suggest that CDKL5 controls flagellar length by modulating IFT. In the absence of CDKL5, flagella have increased levels of IFT proteins and show reduced phosphorylation of the tubulin-binding domain of IFT74, changes that may enhance tubulin transport and promote flagellar elongation. In parallel studies, we found that mouse Cdkl5 is similarly controlled by Cdk20, the mammalian ortholog of LF2, with similar consequences. Our results place CDKL5 as a central regulator of ciliary dynamics, and highlight ciliary dysfunction as a likely driver of pathology in CDD.

## Results

### Phylogeny of CDKL5 and related kinases

The *Chlamydomonas* genome encodes a group of proteins with similarity to the kinase domain of the mammalian CDKLs. In mammals, the CDKL branch of the CDK subfamily is composed of five enzymes that form two clades, one consisting of CDKL5 and the other containing CDKL1, CDKL2, CDKL3, and CDKL4. The *Caenorhabditis* and *Drosophila* genomes each encode a single member, which group most closely to mammalian CDKL1 [9,21–24]. Two of the *Chlamydomonas* CDKLs, CrCDKL5 and CrFAP262, group with mammalian CDKL5 while the remainder form a sister group to the mammalian CDKL1/2/3/4/CeCdkl-1/DmCdkl branch (Fig 1A). Within the CDKL5 group, CrCDKL5 is most closely related to mammalian CDKL5 with the N-terminal kinase domains of CrCDKL5 and HsCDKL5 having similar sequences (S1 Fig), and AlphaFold 3 modeling predicts that they have nearly identical structures (Fig 1B). In CDD patients, missense variants are primarily found in the conserved catalytic domain and are correlated with severe clinical phenotypes [25,26]. Both CrCDKL5 and HsCDKL5 have substantial C-terminal domains with no significant sequence similarity to each other. The other CDKL5-like protein, FAP262, has a catalytic domain similar to CDKL5 but the protein is larger and has its catalytic domain located in the C-terminal end. FAP262 also contains a cluster of IQ motifs and a nucleoside triphosphate binding site not found in CDKL5. Little else is known about FAP262 except that it is a relatively low-abundance ciliary protein [12,27]. The *Chlamydomonas* proteins sister to the CDKL1/2/3/4 branch are mostly uncharacterized but CrFLS1 and CrFLS2 are required for ciliary disassembly [28,29].

Many of the proteins in the CDK and CDKL family have been connected to cilia (asterisks, Fig 1A). We confirmed that mouse CDKL5 localizes to both nonmotile primary cilia and to motile cilia (Figs 1C, 1D, and S2), although we did not observe the prominent enrichment at the proximal end of mammalian cilia that is observed in *Chlamydomonas*.

### Defects in CrCDKL5 alter ciliary waveform

Tam *and colleagues* [2] previously reported that *Chlamydomonas* cells with defects in CDKL5 have long flagella and "move erratically and slowly." In confirmation, we observed abnormally long flagella and erratic slow-swimming in the *lf5-2* allele of *CDKL5* (Fig 2A, 2B, S1, and S2 Videos). This allele removes the 5′ end of the *CDKL5* gene including the exons encoding the kinase domain (S2 Data) and no *CDKL5* message was detected by northern blot [2], suggesting that it is a null allele. Wild-type cells typically swim in straight, loose helical paths. In contrast, *lf5* cells swam in tight helices and

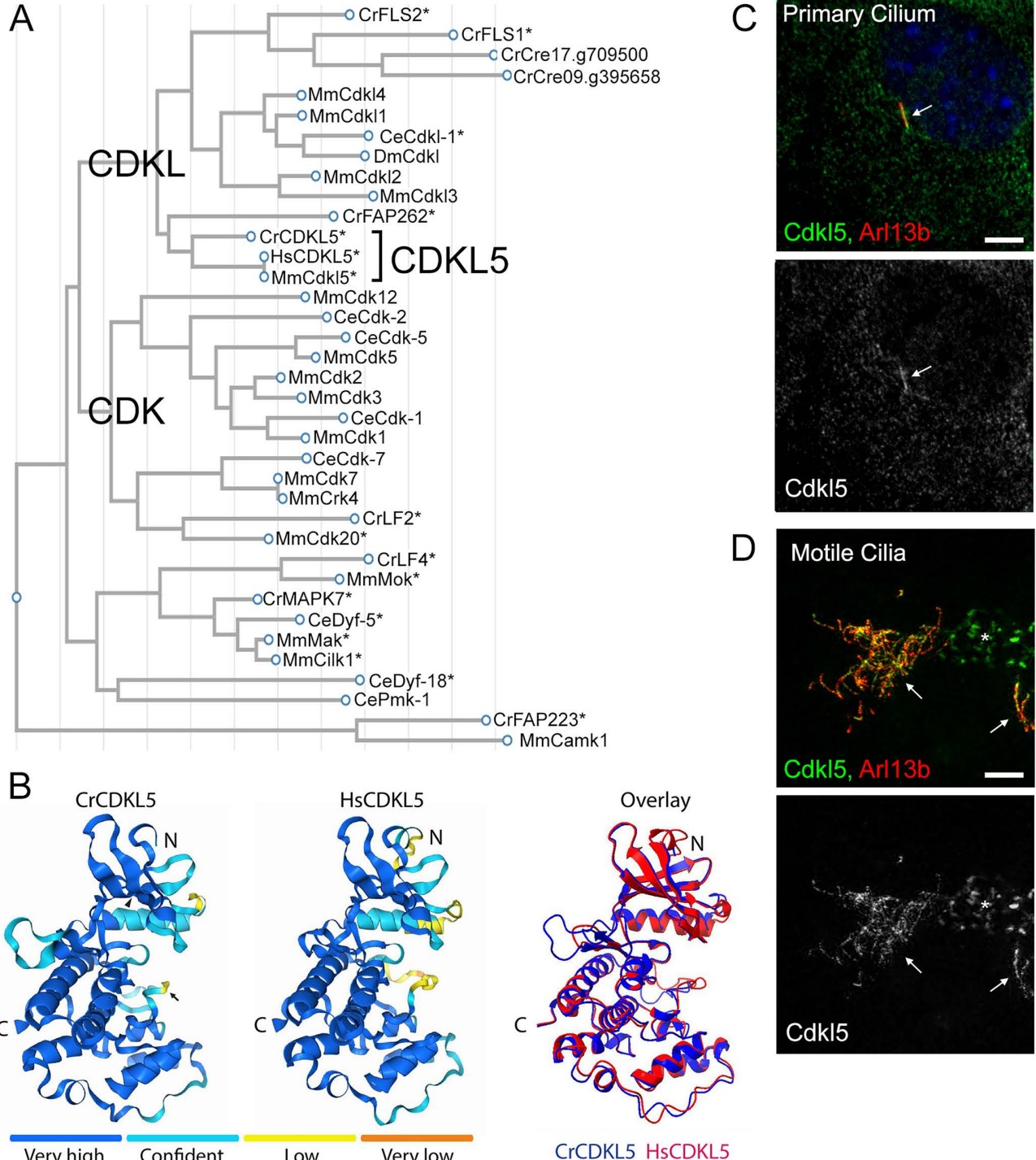

**Fig 1. CrCDKL5 is similar to mammalian CDKL5. (A)** Phylogenetic tree relating the catalytic domains of *Chlamydomonas* CDKL5, human CDKL5, and other CDKL and related kinases from *Chlamydomonas*, mouse, and *Caenorhabditis*. Asterisks (*) mark proteins that have been connected to cilia; see ChlamyFP.org for details. **(B)** AlphaFold 3 models predict similar structures for the N-terminal catalytic domains of CrCDKL5 (amino acids (aa) 1–291, left) and HsCDKL5 (aa 1–295, middle). The models are color coded according to the confidence level as indicated. Very high: pIDDT > 90. Confident: 90 > pIDDT > 70. Low: 70 > pIDDT > 50. Very low: pIDDT < 50. Arrowhead points to K33 and arrow points to Y166 in CrCDKL5. On the right is an overlay of both proteins with CrCDKL5 in blue and HsCDKL5 in red. C, C-terminus. N, N-terminus. **(C, D)** Mouse Cdkl5 localizes to cilia. Cultured mouse

fibroblasts (C) or ependymal cells (D) were stained for endogenous Cdkl5 (green, or gray), Arl13b (red), and DAPI (blue). Z projection of slices taken at 0.24 μm intervals. Scale bars are 5 μm. Arrows mark cilia. * marks Cdkl5 in a nonciliated cell.

altered their swimming direction more frequently. Video analysis showed that swimming speed of *lf5* cells was less than one-half that of wild-type (Fig 2C) and beat frequency of *lf5* cells was reduced to about one-half that of wild-type (Fig 2D). Synchronization of the two flagella was altered in *lf5* cells. In wild-type cells, the two flagella nearly always beat in phase, whereas in *lf5* cells the flagella were out of synchrony in approximately one-half of the observed beat cycles (Fig 2E and S3–S7 Videos). This lack of synchrony is likely a major cause of the frequent turning of *lf5* cells. Flagellar waveform was notably altered in *lf5* cells, with larger bend angles that caused the flagella to frequently cross over to the opposite side of the cell during the beat cycle (Fig 2F). Although the flagella were not observed to become entangled during crossovers, occasionally a tight bend of one flagellum would appear to 'nest' within a tight bend of the other flagellum, perturbing the waveform of one or both flagella (S8 Video). Transformation of *lf5* cells with wild-type CDKL5-GFP fully rescued the long flagella phenotype, almost completely rescued flagellar waveform and beat synchrony, partially rescued swimming speed, but resulted in only a slight increase in beat frequency (Fig 2). The failure to rescue beat frequency is not understood. It is not likely due to low expression of the transgene (S3A Fig) and does not result from the GFP tag on the protein because CDKL5-GFP strains that were created by tagging the endogenous CDKL5 genes in wild-type cells via the CRISPR method, have normal beat frequency (S3B Fig). Our mapping of the deletion that created the *lf5-2* allele revealed that two upstream genes were also deleted (S2 Data). Both are uncharacterized. One is predicted to be a sucrose 6-glucosyltransferase and the other a ribosome-binding GTPase. Neither have been connected to flagella in any proteomic or other study, making their importance to the regulation of beat frequency unknown.

## CDKL5's kinase activity is essential for its function in *Chlamydomonas*

A lysine residue (K33 in *Chlamydomonas* and K42 in mouse) in the CDKL5 catalytic domain coordinates the phosphates of ATP to the C-helix of the active site and is critical for enzymatic activity [30], while a tyrosine residue (Y166 in *Chlamydomonas* and Y171 in mouse and human) in the activation loop controls kinase activity of the human protein [16–19]. To determine if kinase activity is required for CDKL5's function in *Chlamydomonas*, we transformed the *lf5* null mutant with constructs that express enzymatically inactive CDKL5 due to a lysine mutation at the active site (K33R), or CDKL5 in which the activation loop tyrosine was mutated (Y166F). Unlike wild-type CDKL5, the mutant proteins did not restore normal flagellar length or swimming speed (Fig 2A, 2C), showing that CDKL5's kinase activity is essential for regulating length and motility. As expected, given the failure of wild-type CDKL5 to fully rescue beat frequency, the mutant proteins resulted in little (CDKL5[Y166F]) or no (CDKL5[K33R]) rescue (Fig 2D) of this phenotype.

## Mammalian CDKL5 localizes to cilia and defects in it cause abnormally long cilia

While staining mammalian cells with an antibody against endogenous CDKL5 indicated that the protein was enriched in cilia (Figs 1C, 1D, and S2C), the western blot of this antibody recognizes a nonspecific band (S2A Fig). To confirm ciliary localization, we cloned CDKL5 from mouse fibroblasts and tagged it with a 6xMyc tag. Similar to what we observed with the endogenous protein and was previously reported for tagged human protein [9], Myc-tagged protein localized to the basal body region and along the ciliary shaft when expressed in fibroblasts (Fig 3A).

Knockout mouse fibroblast cell lines were obtained from Cdkl5[tm1.1Joez]/J embryos [31], serum starved to induce ciliation and then fixed and stained with cilia markers (Fig 3). This allele has the kinase active site deleted and is expected to be a null allele. In addition, gene editing with three different guide RNAs was used to knock out *Cdkl5* from a MEF line (S4 Fig). Mutant cells derived from the *Cdkl5[tm1.1Joez]* embryos showed a slight but significant increase in cilia length (Fig 3B, 3C). Overexpression of the wild-type gene reduced the cilia length to slightly less than normal. Expression of kinase-dead

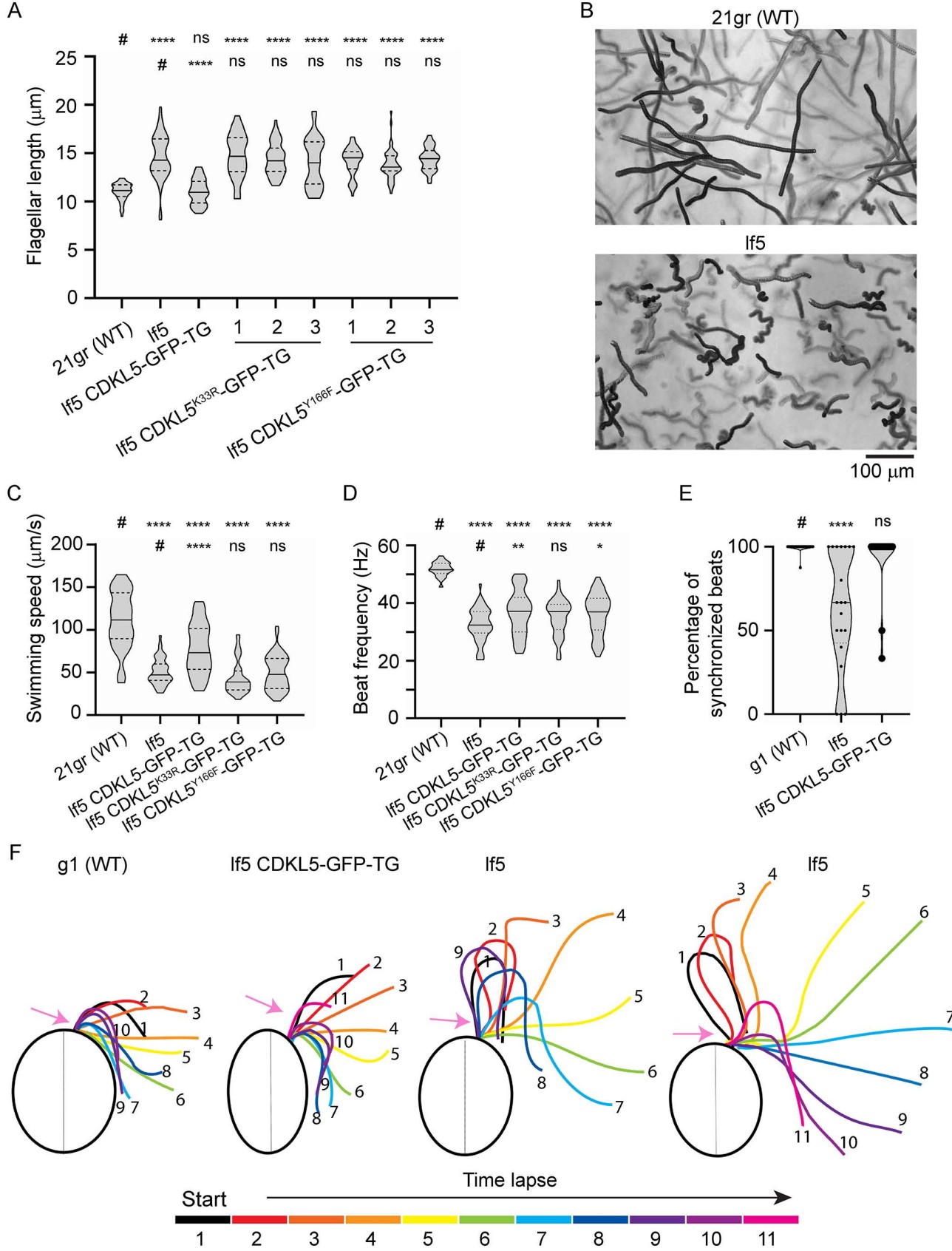

**Fig 2. Defects in CrCDKL5 alter flagellar beat synchrony and waveform.** See Table 1 for a list of all strains and details on them. **(A)** Flagella length of wild-type cells, lf5 cells, and lf5 cell lines expressing wild-type CDKL5-GFP, CDKL5$^{K33R}$-GFP (3 independent cell lines), or CDKL5$^{Y166F}$-GFP (3 independent cell lines). One flagellum from each of 50 cells was measured for each cell line. Wild-type CDKL5-GFP fully rescued the long flagella phenotype of *lf5*, but the mutant CDKL5 proteins did not. ****$p \leq 0.0001$, ns not significant as compared to wild-type (#, top row) or lf5 (#, bottom row) by one-way ANOVA with Tukey's multiple comparisons post-hoc test. Violin plots show median (solid line) and quartiles (dashed lines). Underlying data can be found in S1 Data. **(B)** Swimming paths of wild-type (upper) and lf5 (lower) cells recorded for 2 s. The swimming paths of wild-type cells are longer and straighter than those of lf5 cells, which frequently change direction. **(C)** Swimming speed of wild-type cells, lf5 cells, and lf5 cell lines expressing wild-type CDKL5-GFP, CDKL5$^{K33R}$-GFP, or CDKL5$^{Y166F}$-GFP. lf5 cells swim only about one-half as fast as those of wild-type. CDKL5-GFP partially rescued the slow-swimming phenotype, whereas the mutant CDKL5 proteins did not rescue the phenotype. For each cell line, 50 cells were measured. ****$p \leq 0.0001$, ns: not significant as compared to wild-type (#, top row) or lf5 (#, bottom row) by one-way ANOVA with Tukey's multiple comparisons post-hoc test. Violin plots show median (solid line) and quartiles (dashed lines). Underlying data can be found in S1 Data. **(D)** Beat frequency of wild-type cells, lf5 cells, and lf5 cells expressing wild-type CDKL5-GFP, CDKL5$^{K33R}$-GFP, or CDKL5$^{Y166F}$-GFP. For each cell line, 50 cells were measured. ****$p \leq 0.0001$, **$p \leq 0.01$, *$p \leq 0.05$, ns: not significant as compared to wild-type (#, top row) or lf5 (#, bottom row) by one-way ANOVA with Tukey's multiple comparisons post-hoc test. Violin plots show median (solid line) and quartiles (dashed lines). Underlying data can be found in S1 Data. **(E)** Flagellar synchronization. The percentage of beat cycles where the two flagella were in sync was quantified by viewing movies of swimming cells at slow motion. $n = 20$. ****$p \leq 0.0001$, ns: not significant as compared to wild-type (#) by one-way ANOVA with Tukey's multiple comparisons post-hoc test. Violin plots show means from individual cells with the median (solid line) and quartiles (dashed lines) marked. Underlying data can be found in S1 Data. **(F)** Tracings of flagella from videos of swimming cells of wild-type, lf5, and lf5 rescued with CDKL5-GFP. In each case, tracings were made from 12 equally spaced, sequentially numbered frames of a single beat cycle. Two wild-type, one lf5 CDKL5-GFP-TG, and six lf5 cells were examined. Traces from wild-type and lf5 CDKL5-GFP-TG were similar and one example of each is shown. lf5 traces were variable, and so two examples are shown. In lf5, the principal bend angle of the recovery stroke is much greater than that in wild-type and rescue and there is a larger reverse bend near the base of the flagellum (arrows, color coded to match tracing). As a result, the flagella of lf5 often cross over to the opposite side of the cell.

Cdkl5$^{K42A}$ did not restore cilia length indicating that kinase activity is fundamental to Cdkl5's function in regulating cilia length. Expression of Cdkl5$^{T169A,Y171F}$ with the potential activation loop phosphorylation sites mutated was able to partially rescue although not as well as the wild-type construct. (Fig 3B, 3C). CRISPR knockout cells from each of the guides also had longer cilia that could be rescued with a CRISPR-resistant wild-type construct (S4 Fig).

## CDKL5 is extensively phosphorylated, including at three sites within the activation loop

Phosphorylation of human CDKL5 on tyrosine residue 171 within the activation loop is thought to control CDKL5's kinase activity [16–19]. Our data indicated that the equivalent residue in *Chlamydomonas* CDKL5 (Y166) is critical to the protein's function, raising the question of whether control of CrCDKL5 is mediated in part by phosphorylation of Y166. To determine the phosphorylation profile of CrCDKL5, we purified CDKL5-GFP and analyzed it by mass spectrometry (MS). In six independent experiments, we found a total of 47 sites were phosphorylated (Fig 4A and S3 Data: sheet "CrCDKL5 phosphorylation sites"). The residues of the activation loop (amino acids [aa] 149–176) were of particular interest. Most of the loop is contained in one tryptic peptide covering aa 155–170, which has five potentially phosphorylatable residues at S162, T164, Y166, S168, and T169. S162 was phosphorylated in three experiments, T164 in 4 experiments, and Y166 in 4 experiments. A closer look at this peptide revealed phosphoisoforms mono-phosphorylated, bi-phosphorylated, and tri-phosphorylated at combinations of these three residues, as well as the unphosphorylated isoform (S3 Data, sheet "Peptide 155-170"). Within the broader catalytic domain (aa 1–291), phosphorylation was detected on residues Y4 (3 experiments), S8 (two experiments), S135 (one experiment), and Y184 (two experiments). Numerous other phosphorylation sites were detected in the C-terminal half of CDKL5, including S584 (4 experiments), which will be discussed later. In total, 23% (7 out of 31) of potential sites were phosphorylated in the N-terminal kinase domain, whereas a remarkable 80% of candidate sites (40 out of 50) were phosphorylated in the C-terminal domain (Fig 4A).

## CrCDKL5 is phosphorylated by LF2

In addition to CDKL5, the *Chlamydomonas* long flagella genes *LF2* and *LF4* encode protein kinases. LF2 encodes a conserved member of the CDK family, known as Cdk20 in mammals, and LF4 encodes a conserved member of the mitogen-activated protein kinase family, known as MOK in mammals [20,32,33] (Fig 1A). To investigate if these kinases

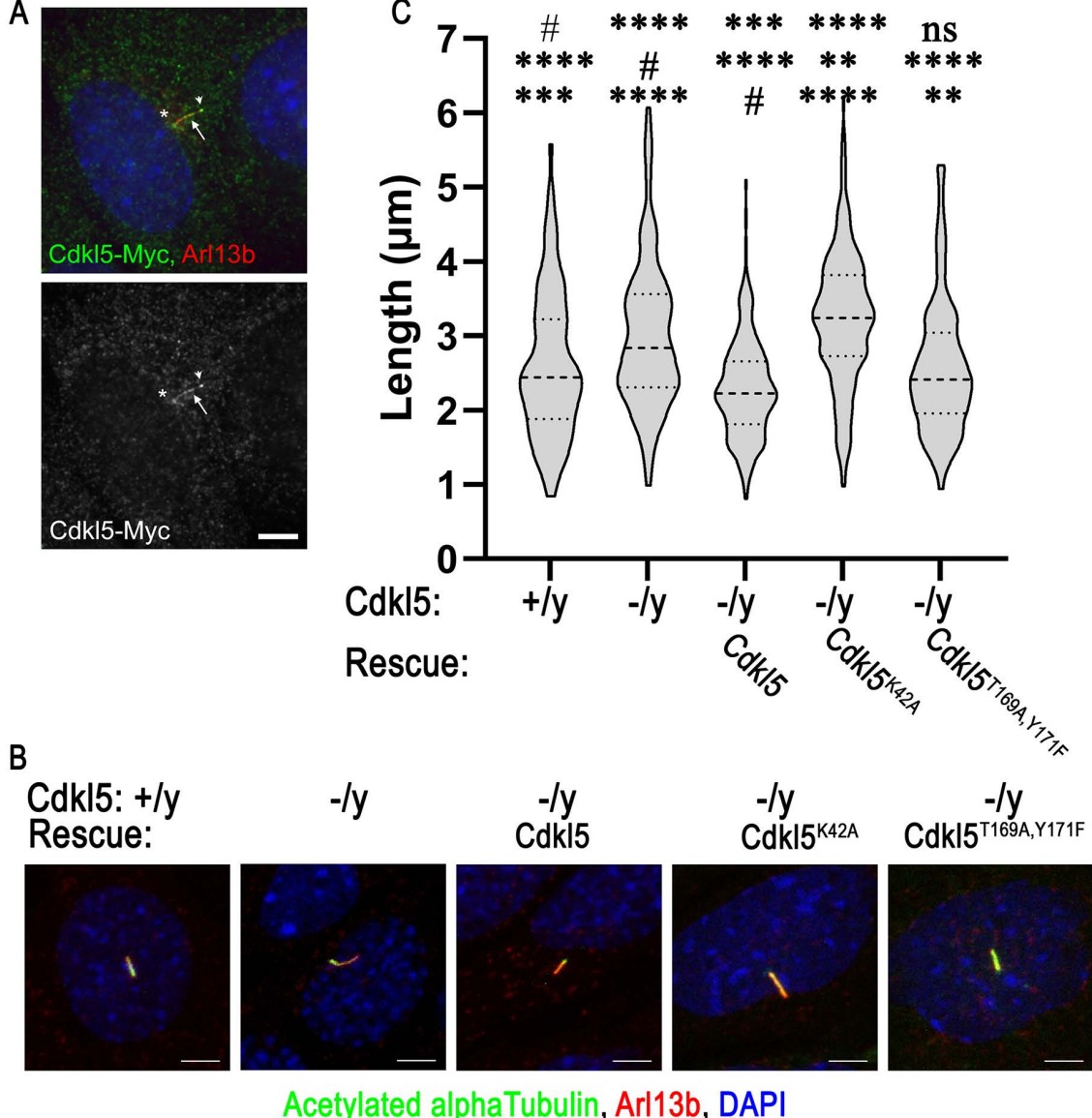

**Fig 3. CDKL5 ciliary function is conserved in mouse. (A)** Mouse embryonic fibroblast (MEF) labeled for Cdkl5-6xMyc (Myc, green), cilia (Arl13b, red), and DNA (DAPI, blue). Scale bar, 5 μm. Z projection of slices taken at 0.24 μm intervals. **(B)** MEFs derived from a control male embryo (+/y) and a hemizygous mutant male embryo (−/y), along with mutant MEFs transfected with DNA constructs expressing wild-type (Cdkl5-6xMyc), kinase-dead (Cdkl5$^{K42A}$-6xMyc), or activation loop-mutated (Cdkl5$^{T169A,Y171F}$-6xMyc) Cdkl5 were stained for cilia (acetylated tubulin in green, Arl13b in red) and DAPI (blue). Scale bar, 5 μm. Z projection of slices taken at 1-μm intervals. See S3 Table for details of the plasmid constructs. **(C)** Quantification of cilia length in the cells described in B. ****$p \le 0.0001$, ***$p \le 0.001$, ns: not significant as compared to control (#) by one-way ANOVA with Tukey's multiple comparisons post-hoc test. $n > 200$ for each condition. Violin plots show median (darker dashed line) and quartiles (dashed lines). Underlying data can be found in S1 Data.

phosphorylate CDKL5, we tagged endogenous CDKL5 with GFP via CRISPR in wild-type, *lf2* mutant, and *lf4* mutant cells. The *lf2-5* allele used for our experiments is predicted to express a C-terminally truncated LF2 protein lacking part of the kinase domain but retaining some function [20] while the *lf4-9* allele we used is likely null [2]. We then isolated the flagella and compared the migration patterns of their CDKL5-GFP using Phos-tag SDS-PAGE (Fig 4B), which has the ability to

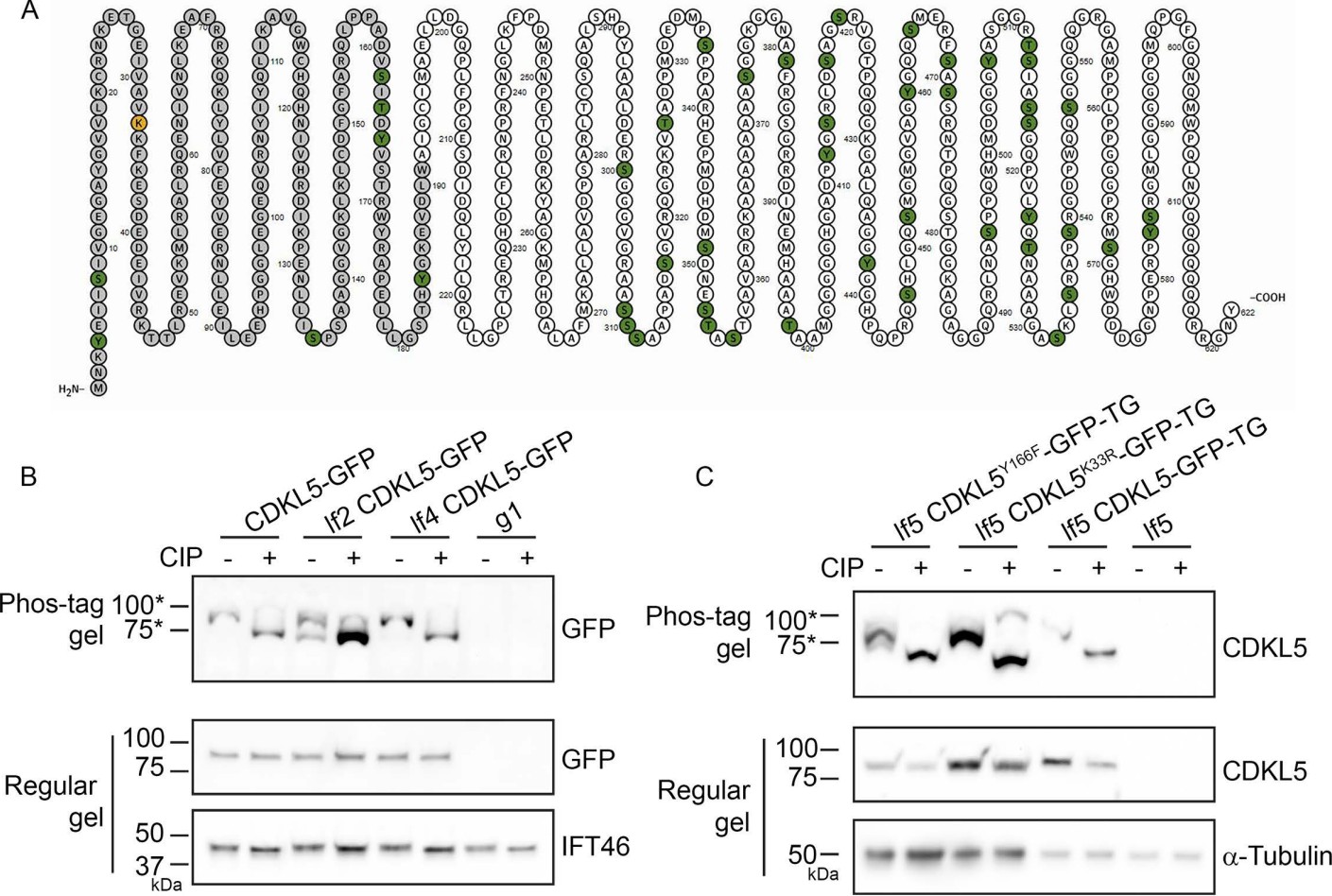

**Fig 4. LF2 phosphorylates CrCDKL5. (A)** Diagram of phosphorylation sites (green residues) and active site lysine (orange residue) in CDKL5. The residues of the conserved catalytic domain are darker gray. Diagram drawn with Protter [89]. **(B)** Flagella samples prepared from CDKL5-GFP cells, lf2 CDKL5-GFP cells, and lf4 CDKL5-GFP cells expressing GFP-tagged CrCDKL5 in the absence of untagged CDKL5 (see Table 1 for details of all strains used) were treated either with or without calf-intestinal phosphatase (CIP). The samples were electrophoresed on a Phos-tag SDS gel (upper panel) or a regular SDS gel (middle panel) and probed with anti-GFP antibody to detect GFP-tagged CrCDKL5. A sample of flagella from wild-type cells (g1) containing only untagged CrCDKL5 was treated and loaded the same way to serve as a control for antibody specificity. The same samples were also electrophoresed in a regular SDS gel and probed with anti-IFT46 antibody as a loading control (lower panel). CDKL5-GFP from both CDKL5-GFP and lf4 CDKL5-GFP flagella without CIP treatment migrated as a single band above the 75-kD marker. Following CIP treatment, most of the CDKL5-GFP in these samples shifted to a new, faster migrating band. In contrast, CDKL5-GFP from lf2 CDKL5-GFP flagella without CIP treatment migrates as two more-or-less equally dark bands at positions corresponding to those of CIP-treated and untreated CDKL5-GFP from flagella of CDKL5-GFP cells. After CIP treatment nearly all of the protein shifts to the faster moving band. These results indicate that normal phosphorylation of CrCDKL5 requires LF2. One example of 6 repeats is shown. **(C)** Flagella samples prepared from lf5 mutant cells expressing CDKL5$^{Y166F}$-GFP, CDKL5$^{K33R}$-GFP, and CDKL5-GFP were treated either with or without CIP. The samples were separated by Phos-tag SDS-PAGE (upper panel) or regular SDS-PAGE (middle panel) and probed with anti-CrCDKL5 antibody. A sample of lf5 flagella was treated and loaded the same way as a control for antibody specificity. The same samples were also separated on a regular SDS gel and probed with an α-tubulin antibody as a loading control (lower panel). No obvious migration differences were detected between any of the samples. One example of two repeats is shown. *On a Phos-tag SDS gel, protein migration is not strongly correlated with mass.

separate different isoforms based on their phosphorylation states [34,35]. CDKL5-GFP from wild-type and *lf4* flagella migrated as a single band above the 75-kD marker. In both cases, treatment with calf-intestinal phosphatase (CIP) yielded a new, faster migrating band presumably corresponding to a relatively unphosphorylated form of CDKL5-GFP. In contrast,

CDKL5-GFP from *lf2* flagella migrated at both positions prior to phosphatase treatment, with nearly all protein being in the faster migrating band after the treatment. This analysis suggests that CDKL5 is phosphorylated in both wild-type and *lf4* flagella and that the LF2 kinase is responsible for at least a portion of CDKL5's phosphorylation.

The activity of mammalian CDKL5 is thought to be controlled by autophosphorylation [16–19]. To determine if the phosphorylation of CrCDKL5 is dependent upon the protein's own catalytic activity, we compared the Phos-tag gel migration patterns of flagellar CDKL5-GFP, CDKL5$^{Y166F}$-GFP, and CDKL5$^{K33R}$-GFP proteins expressed in a *lf5*-null background (Fig 4C). In each case, the untreated CDKL5 variants migrated as a single band above the 75-kD marker, whereas all or nearly all CIP-treated CDKL5 variants ran as a single band below the marker indicating that autophosphorylation is not a major contributor to the phosphorylation of CrCDKL5.

## Phosphorylation at the CrCDKL5 activation loop is driven by LF2

To confirm that CDKL5 is a substrate of LF2 and to identify the LF2 phosphorylation sites on CDKL5, we purified CDKL5-GFP from whole-cell lysates of lf5 CDKL5-GFP-TG cells and lf2 CDKL5-GFP-TG cells along with CDKL5$^{K33R}$-GFP from lf5 CDKL5$^{K33R}$-GFP-TG cells in two independent experiments (details of strains used are in Table 1). We then analyzed the samples by MS with attention to precursor intensities, a value frequently used for quantification in label-free MS [36]. For the most part, the phosphorylation pattern of CDKL5 was similar in all three strains (S4 Data). Peptide abundances were generally lower for phosphorylated peptides than for nonphosphorylated peptides, and often near the limit of detection, so that many phosphoisoforms were detected at levels too low to quantitate or were detected in one experiment. Despite these issues, there was a clear anomaly for the activation loop peptide (aa 155–170, highlighted in yellow in S4 Data). Unexpectedly, we found that the nonphosphorylated isoform was very abundant in CDKL5 purified from *lf2* cells but much less abundant in the samples from *lf5* cells expressing wild-type or kinase-dead CDKL5 (Fig 5A). Comparing the ratios of the sum of the precursor intensities for each individual CDKL5 peptide from *lf2* versus wild-type samples and from *lf2* versus CDKL5$^{K33R}$ samples (Fig 5B) shows that for most peptides the ratios were close to 1, indicating that peptides were of similar abundance in the two samples. However, the activation loop peptide was more than 100 fold more abundant in CDKL5 purified from *lf2* mutant cells as compared to wild-type CDKL5 or kinase-dead CDKL5 purified from *LF2* wild-type cells.

The above findings presented a conundrum—why were the activation loop peptides not being detected in samples from control cells expressing CDKL5-GFP or CDKL5$^{K33R}$-GFP even though they were readily detected when the protein was obtained from *lf2* cells? One possibility is that the LF2 kinase phosphorylates CDKL5 at a position that blocks trypsin cleavage so that the activation loop peptide is not released from adjacent peptides. If this were the case, we would expect a longer peptide to be generated. R170 next to S168 and T169 would be the most likely blocked cleavage site. Failure to cleave at R170 would produce a peptide from aa 155–173 but this peptide was not detected (S2 Table). Similarly, there was no evidence for a missed cleavage involving R154 at the N-terminus of the activation loop peptide. Therefore, failure to be cleaved is unlikely to account for the reduced amount of activation loop peptide detected in CDKL5 or CDKL5$^{K33R}$ expressed in the presence of LF2.

Another possibility is that a strong negative charge resulting from phosphorylation of three or more residues within the activation loop peptide prevented its detection. To test this hypothesis, we isolated wild-type and kinase-dead CDKL5 expressed in the presence of LF2, and treated a portion of each sample with CIP before MS analysis. CIP treatment greatly elevated the amount of mono-phosphorylated and unphosphorylated activation loop peptide in both samples (Fig 5C). Again, ratioing the sum of the precursor intensities for each individual CDKL5 peptide showed that most peptides had a ratio of about one indicating that the same amount of peptide was found in both the CIP-treated and untreated samples. However, peptide aa155–170 covering the activation loop was >90 fold more abundant in the CIP-treated samples (Fig 5D). These data strongly support the hypothesis that LF2 is responsible for multi-site phosphorylation of CDKL5 on S162, T164, and Y166 within the activation loop and that most copies of the LF5 protein are normally phosphorylated at

PLOS Biology

**Table 1.** *Chlamydomonas* strains.

| Strain name | CRC number[*] | Genotype[#] | Reference | Notes |
|---|---|---|---|---|
| g1 | CC-5415 | *nit1; agg1; mt+* | [86] | Wild-type; high-efficiency transformation |
| 21gr | CC-1690 | *mt+* | [87] | Wild-type |
| 137c | CC-125 | *nit1; nit2; mt+* | [87] | Wild-type |
| lf2 | CC-2287 | *lf2-5; mt−* | [88] | *lf2-5* is a hypomorphic allele |
| lf5 | CC-4560 | *lf5-2; mt−[§]* | [2] | Tam DKD6 strain. Deletion includes the kinase domain (S2 Data). No *CDKL5* message was detected by northern blot [2]. |
| lf4 | CC-4768 | *lf4-9; mt+* | [2] | Tam D12 strain [2]. |
| CDKL5-HA | CC-5971 | *CDKL5:HA; nit1; agg1; mt+* | [70] | Made by TIM-tagging in g1 wild-type (CC-5415) |
| Wild-type mt+ | CC-620 | *nit1, nit2, mt+* | | Wild-type, high-efficiency mating, for autolysin preparation |
| Wild-type mt− | CC-621 | *nit1, nit2, mt−* | | Wild-type; high-efficiency mating, for autolysin preparation |
| fap93 | CC-6020 | *fap93−1, mt+* | [44] | |
| fap93 GFP-AP93-HA-TG | CC-6021 | *fap93−1; GFP:FAP93:HA-TG* | [44] | |
| CDKL5-GFP (T6A4[$]) | CC-6302 | *CDKL5:GFP nit1; agg1; mt+* | (this work) | Made by TIM-tagging in g1 wild-type (CC-5415) with LF5 gRNA-1 and 3751 bp PCR product amplified from pLF5CsfGFP using primer pair LF5donor-1/LF5donor-2 |
| lf2 CDKL5-GFP (T1C2[$]) | CC-6303 | *lf2-5; CDKL5:GFP; mt−* | (this work) | Made by TIM-tagging in lf2 (CC-2287) with LF5 gRNA-1 and 3751 bp PCR product amplified from pLF5CsfGFP using primer pair LF5donor-1/LF5donor-2 |
| lf4 CDKL5-GFP (T3D2[$]) | CC-6304 | *lf4-9; CDKL5:GFP; mt+* | (this work) | Made by TIM-tagging in lf4 (CC-4768) with LF5 gRNA-1 and 4096 bp PCR product amplified from pLF5CsfGFPHyg using primer pair LF5donor-1/Hyg-4 |
| lf5 CDKL5-GFP-TG (C12[$]) | CC-6305 | *lf5-2; CDKL5:GFP-TG; mt−* | (this work) | Made by transformation of lf5 (CC-4560) with ScaI linearized pLF5CsfGFP plasmid |
| lf5 CDKL5$^{K33R}$-GFP-TG (K33R1[$]) | CC-6306 | *lf5-2; CDKL5$^{K33R}$:GFP-TG; mt−* | (this work) | Made by transformation of lf5 (CC-4560) with ScaI linearized pLF5-K33R plasmid |
| lf5 CDKL5$^{Y166F}$-GFP-TG (Y166F1[$]) | CC-6307 | *lf5-2; CDKL5$^{Y166F}$:GFP-TG; mt−* | (this work) | Made by transformation of lf5 (CC-4560) with ScaI linearized pLF5-Y166F plasmid |
| lf5 CDKL5$^{S162A,T164A,Y166A}$-GFP-TG (3AB5[$]) | CC-6308 | *lf5-2; CDKL5$^{S162A,T164A,Y166A}$:GFP-TG; mt−* | (this work) | Made by transformation of lf5 (CC-4560) with ScaI linearized pLF5-S162A,T164A,Y166A plasmid |
| g1 CDKL5-GFP-TG (g1CDKL5F1[$]) | CC-6309 | *CDKL5:GFP-TG; mt+* | (this work) | Made by transformation of g1 wild-type (CC-5415) with ScaI linearized pLF5CsfGFP plasmid |
| g1 CDKL5$^{K33R}$-GFP-TG (g1K33RA2[$]) | CC-6310 | *CDKL5$^{K33R}$:GFP-TG; mt+* | (this work) | Made by transformation of g1 wild-type (CC-5415) with ScaI linearized pLF5K33R plasmid |

[*] CC strains are available from the *Chlamydomonas* Resource Center (www.chlamycollection.org).

[§] CC-4560 is reported to be *mt+* but our analysis indicates that it is *mt-*.

[#] -TG indicates a transgene inserted at random site in the genome.

[$] Witman lab IDs.

these sites. The results for CDKL5$^{K33R}$ further show that the phosphorylation of these sites is not dependent on CDKL5 autophosphorylation.

The quantitative comparison of phosphopeptides in the three samples also revealed phosphorylation sites of potential interest in the C-terminal domain of CDKL5 (S4 Data). S584, which centers on a phosphorylation-consensus sequence for CDKL5 (see subsection "Global effects on flagellar composition and protein phosphorylation due to the lack of CDKL5"),

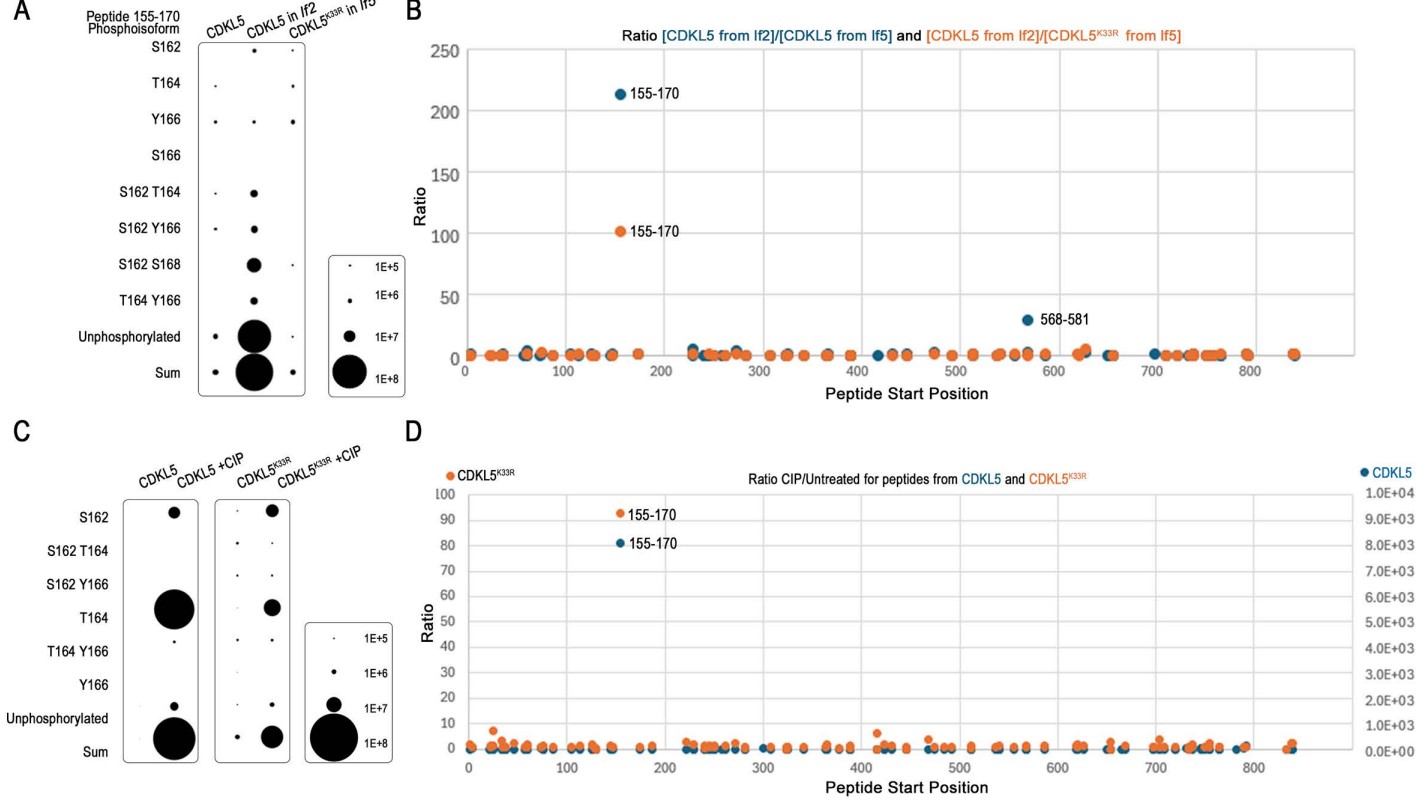

**Fig 5. MS analysis of CDKL5 activation loop phosphopeptides. (A)** Average abundance (precursor intensity) of the phosphoisoforms of the activation loop peptide (amino acids (aa) 155–170) in wild-type CDKL5, CDKL5 expressed in *If2* cells, and CDKL5$^{K33R}$ expressed in *If5* cells. The scale correlating dot size to abundance is in the box at the bottom right of the panel. Underlying data can be found in S1 Data. **(B)** The sum of the precursor intensities for each individual CDKl5 peptide was ratioed from protein purified from *If2* cells (If2 CDKL5-GFP-TG) as compared to protein purified from *If5* cells (If5 CDKL5-GFP-TG or If5 CDKL5$^{K33R}$-GFP) cells. A ratio near 1 indicates similar abundance in both samples, whereas higher numbers indicate enrichment in protein purified from *If2* cells. Underlying data can be found in S4 Data on sheet "All CDKL5 peptides." **(C)** Abundance (precursor intensity) of the phosphoisoforms of the activation loop peptide (aa 155–170) in wild-type CDKL5 and kinase-dead CDKL5$^{K33R}$ without and with phosphatase treatment (+CIP). The scale correlating dot size to abundance is in the box at the bottom right of the panel. Note that all untreated wild-type CDKL5 peptides were less than 1E+5 in abundance and thus are not visible at this scale. Underlying data can be found in S1 Data. **(D)** The sum of the precursor intensities for each individual CDKL5 peptide was ratioed from CIP-treated compared to untreated. A ratio near 1 indicates similar abundance in both samples, whereas higher numbers indicate enrichment in protein treated with phosphatase. Note that the scales for the two experiments are different. Underlying data can be found in S4 Data on sheet "CDKL5 (WT and K33R) CIP peptides".

was phosphorylated in wild-type CDKL5 but not in kinase-dead CDKL5$^{K33R}$ or CDKL5 from *If2* cells, indicating that the site is auto-phosphorylated in an LF2-dependent manner. A similar pattern involving mono- and bi-phosphorylation at Y412, S418, and S421 in CDKL5-GFP from wild-type cells but not from the other two samples suggests that phosphorylation at these sites is dependent on the activity of both CDKL5 and LF2, although residues surrounding these sites do not match the CDKL5 consensus sequence.

## LF2 is required for CDKL5's kinase activity

Several studies have demonstrated autophosphorylation activity of mammalian CDKL5, which is thought to be occurring on the tyrosine of the activation loop [16–18]. Our results indicate that the *Chlamydomonas* CDKL5 activation loop is phosphorylated by LF2 rather than by autophosphorylation but we identified other residues as potential sites for

autophosphorylation. To determine if CrCDKL5 is capable of autophosphorylation, CDKL5-GFP, CDKL5$^{K33R}$-GFP, and CDKL5$^{S162A,T164A,Y166A}$-GFP in which the three phosphorylated residues within the activation loop were mutated to alanine, were immunoprecipitated from *lf5* cells*, incubated with and without adenosine-5′-O-(3-thiotriphosphate) (ATPγS), and then analyzed by western blotting using a thiophosphate-specific antibody. Thiophosphorylation was readily detected on CDKL5 but not on CDKL5$^{K33R}$ (Fig 6) indicating that CDKL5 can autophosphorylate. No thiophosphorylation was detected on CDKL5$^{S162A,T164A,Y166A}$, likely indicating that the kinase is unable to be activated, but this experiment cannot rule out that one or more of these residues is the target of thiophosphorylation.

To determine if CDKL5 autophosphorylation is dependent on the protein kinases LF2 or LF4, wild-type CDKL5-GFP was isolated from cells of the *lf2* or *lf4* mutants and analyzed as above. Little or no thiophosphorylation was detected on CDKL5 isolated from *lf2* cells but was readily detected on CDKL5 isolated from *lf4* cells (Fig 6), indicating that CDKL5's ability to autophosphorylate is dependent on LF2 but not LF4.

## Normal distribution of CrCDKL5 within flagella requires its kinase activity and is dependent on LF2

While it has long been known that CDKL5 is present in the flagellum [2,12], the relative distribution of the protein between cell body and flagella, and whether this distribution requires the protein to be catalytically active, is unknown. To investigate this, we separated cell bodies and flagella from cells expressing CDKL5-HA, CDKL5$^{K33R}$-GFP, and CDKL5$^{Y166F}$-GFP and compared the samples by western blotting. We found that the majority (>80%) of CDKL5-HA is localized to the flagella with only a small amount in the cell body (Fig 7A). CDKL5$^{K33R}$-GFP and CDKL5$^{Y166F}$-GFP were similarly enriched in flagella compared to cell bodies (Fig 7A), indicating that CDKL5 transport into flagella does not require its kinase activity. There was less mutant CDKL5 than wild-type CDKL5 in both whole-cell and flagella samples (Fig 7B), suggesting that the mutant CDKL5 is less stable than wild-type.

To determine how CDKL5's kinase activity and phosphorylation of the activation loop affects CDKL5's distribution in flagella, we used immunofluorescence microscopy to compare *lf5* cells expressing CDKL5-GFP, CDKL5$^{K33R}$-GFP, CDKL5$^{Y166F}$-GFP, or CDKL5$^{S162A,T164A,Y166A}$-GFP. Untransformed *lf5* cells were used as a control for cellular autofluorescence. Consistent with the observations of Tam and colleagues for endogenous CDKL5 [2], wild-type CDKL5-GFP was highly concentrated at the base of steady-state flagella with a few spots along the shaft. In contrast, CDKL5$^{K33R}$-GFP and CDKL5$^{S162A,T164A,Y166A}$-GFP were spread along the flagellar shaft without concentrating at the base. CDKL5$^{Y166F}$-GFP was significantly dispersed from the flagella base but still showed a slight enrichment at this site (Fig 7C, 7D). CDKL5-GFP expressed in *lf2* mutant cells distributed similarly to CDKL5$^{Y166F}$-GFP (Fig 7C, 7D) consistent with a role for LF2 in phosphorylation of CDKL5's activation loop. Additionally, kinase-dead CDKL5$^{K33R}$-GFP expressed in a wild-type background (i.e., in the presence of endogenous CDKL5) had a distribution similar to that of CDKL5$^{K33R}$-GFP expressed in a *lf5* background (S5 Fig), indicating that the CDKL5 kinase activity cannot be supplied in trans to restore normal localization of the protein.

## Normal localization of mammalian CDKL5 requires Cdk20

Mouse Cdkl5 is normally localized along the ciliary shaft with an additional pool in the basal body region outside of the cilium (Figs 1C and 3A). To determine if mammalian Cdkl's ciliary localization is regulated like *Chlamydomonas* CDKL5, we compared the ciliary distribution of wild-type Cdkl5, Cdkl5 in which the active site was disrupted (Cdkl5$^{K42A}$), and Cdkl5 in which the potential activation loop phosphorylation sites were mutated (Cdkl5$^{T169A,Y171F}$). Expressing the 6xMyc-tagged constructs in *Cdkl5* mutant fibroblasts showed that, compared to cells rescued with wild-type, the tip localization was slightly increased when the activation loop was mutated, and the base enrichment was decreased for proteins with either the kinase-dead or activation loop mutations (Fig 8). These results were similar to, albeit less striking, than those seen with equivalent mutant forms of CrCDKL5 in *Chlamydomonas*.

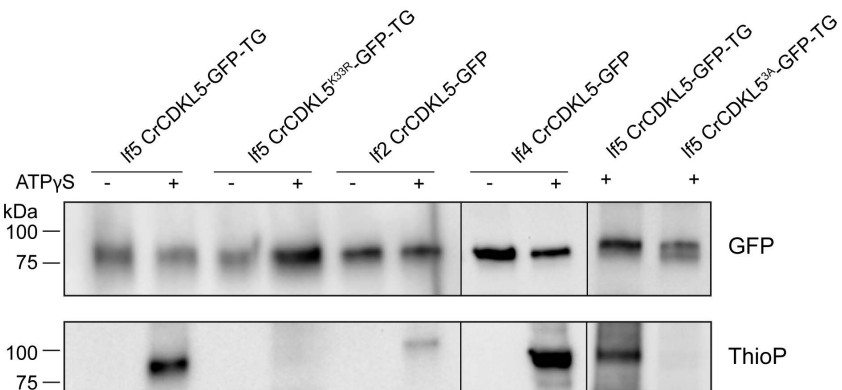

**Fig 6. LF2 kinase activity is required for CDKL5 autophosphorylation ability.** CDKL5-GFP, kinase-dead CDKL5$^{K33R}$-GFP, and CDKL5$^{S162A,T164A,Y166A}$-GFP were immunopurified from *lf5* mutant cells along with CDKL5-GFP from *lf2* and *lf4* mutant cells. The extracts were incubated with or without adenosine-5′-O-(3-thiotriphosphate) (ATPγS) and analyzed by western blot with antibodies to GFP (top panel) or thiophosphate (bottom panel). Example shown is one of three repeats except for CDKL5$^{S162A,T164A,Y166A}$-GFP and lf4 CDKL5-GFP, which were done once. CDKL5-GFP expressed in *lf5* or *lf4* mutant cells can auto-thiophosphorylate in vitro, whereas little or no auto-thiophosphorylation was detected for CDKL5$^{K33R}$-GFP or CDKL5$^{S162A,T164A,Y166A}$-GFP from lf5 cells, or CDKL5-GFP from lf2 cells.

The mammalian ortholog of *Chlamydomonas* LF2 is thought to be Cdk20 [37]. To test the involvement of Cdk20 in regulating Cdkl5 localization, we removed *Cdk20* from *Cdkl5*-mutant fibroblasts (S6 Fig) and rescued the *Cdkl5* mutation with the Cdkl5-6xMyc, Cdkl5$^{K42A}$-6xMyc, and Cdkl5$^{T169A,Y171F}$-6xMyc constructs (Fig 8). The loss of *Cdk20* increased cilia length more than the loss of *Cdkl5*. Comparing cilia lengths in mutant cells rescued with wild-type *Ckdl5*, *Cdkl5*-kinase-dead, or a *Cdkl5* activation loop mutant indicates that the loss of both *Cdkl5* and *Cdk20* does not have an additive effect on cilia length, consistent with them working in the same pathway. Compared to Cdk20 wild-type cells, the loss of Cdk20 shifted wild-type CDKL5 and both mutant CDKL5s from the basal body region into the cilium, where they concentrated near the tip, indicating that Cdk20 has a role similar to that of *Chlamydomonas* LF2 in determining the normal localization of CDKL5. Interestingly, the loss of Cdk20 had a greater effect on both ciliary length control and CDKL5 distribution than the loss of Cdkl5 kinase activity, suggesting that Cdk20 has additional roles beyond activating CDKL5's kinase activity.

## Localization and intraflagellar trafficking of CDKL5 in vivo

CDKL5 accumulates in flagella, where it undergoes dynamic redistribution as growing flagella approach steady-state [2]. To determine if CDKL5 is moved within flagella by IFT, we used total internal reflection fluorescence (TIRF) microscopy to investigate the movement of wild-type CDKL5-GFP in living *lf5* mutant cells. In regenerating flagella (Fig 9A, top), CDKL5 exhibited movement typical of IFT, with numerous anterograde events observed. Many of these events were highly processive—i.e., individual transport events continued uninterrupted to the flagellar tip where CDKL5 accumulated. Far fewer retrograde movements were observed. A smaller pool of CDKL5 was observed at the base of the flagellar shaft. Our studies did not distinguish if this pool represented protein entering the flagellum or returning from the tip. In contrast, in steady-state flagella (Fig 9A, bottom), most of the CDKL5 was located at the base of the flagellar shaft rather than the tip, anterograde movements were fewer (Fig 9D) and less processive, and more diffusion was apparent. We conclude that in growing flagella, CDKL5 is actively moved towards the tip by IFT whereas retrograde movement is by diffusion, and that IFT of CDKL5 is down-regulated when the flagellum reaches steady-state.

To determine if IFT of CDKL5 is dependent on its kinase activity, we observed steady-state flagella of *lf5* cells expressing CDKL5$^{K33R}$-GFP, CDKL5$^{Y166F}$-GFP, or CDKL5$^{S162A,T164A,Y166A}$-GFP (Fig 9B, 9C, 9E). Compared to wild-type, anterograde IFT of kinase-dead CDKL5$^{K33R}$ was highly upregulated in steady-state flagella (Fig 9F). Consistent with analysis of stained

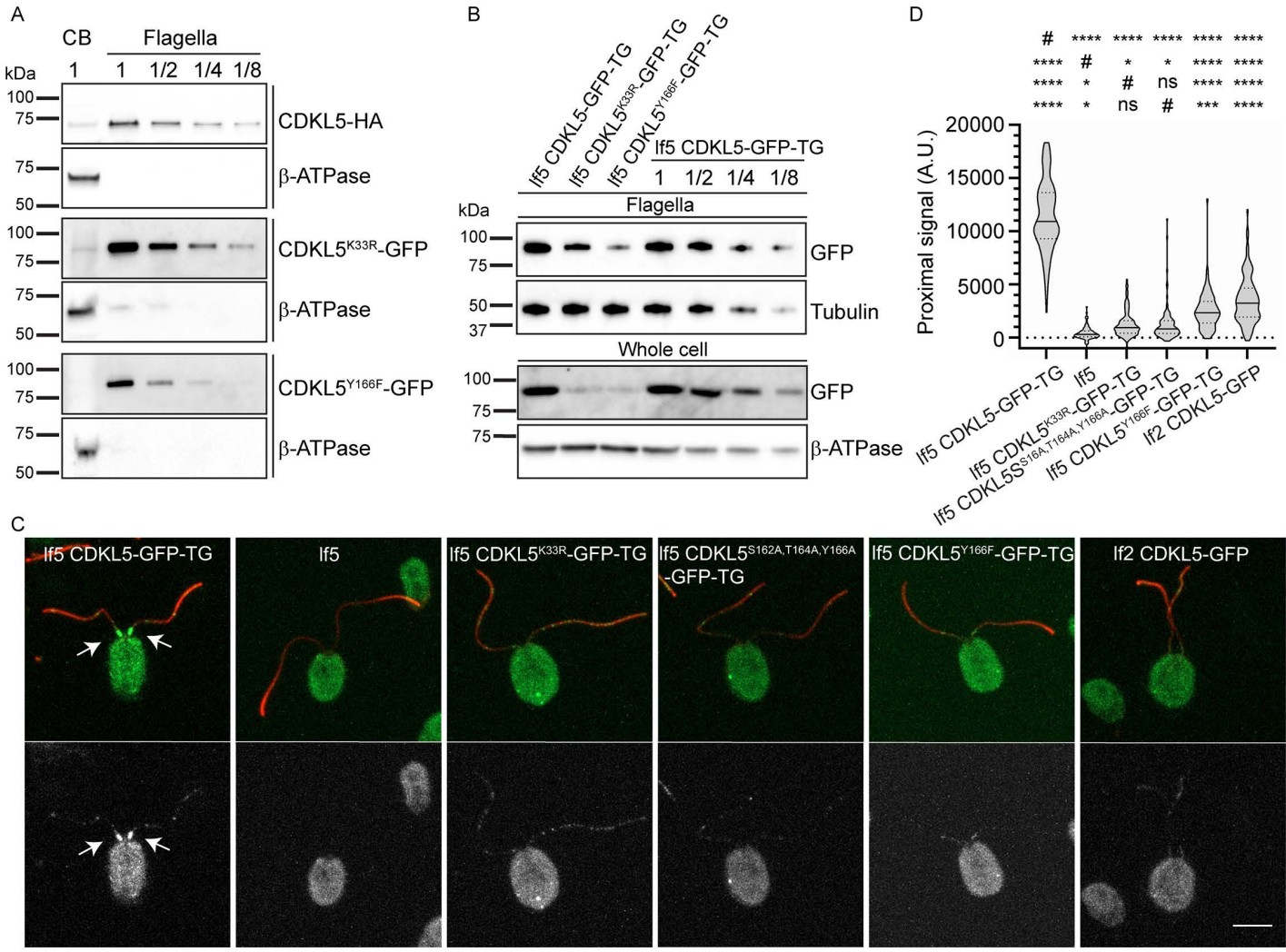

**Fig 7. CDKL5's distribution within the flagellum is dependent on its kinase activity. (A)** Like their wild-type counterpart, mutant CDKL5 proteins mainly localize in flagella. Western blots of cell body (CB) and flagella samples from cells expressing CDKL5-HA, CDKL5^K33R-GFP, or CDKL5^Y166F-GFP were probed with anti-HA or anti-GFP to reveal CDKL5 proteins, or with anti-β-ATPase, which is located only in the cell body, as a control. Numbers indicate the relative amounts of cell bodies and flagella loaded on the gel, with "1" indicating one cell body or two flagella. **(B)** Protein levels of mutant CDKL5s are lower than those of wild-type CDKL5 in both whole-cell and flagella samples. Western blots of flagella (upper panels) and whole-cell (lower panels) samples from lf5 cells expressing CDKL5-GFP, CDKL5^K33R-GFP, or CDKL5^Y166F-GFP were probed with anti-GFP to show CDKL5 proteins, or with anti-α-tubulin or anti-β-ATPase as controls. Samples from lf5 cells expressing CDKL5-GFP were loaded in a dilution series to facilitate comparison. Details for strains used are in Table 1. **(C)** lf5 mutant cells untransformed, or expressing CDKL5-GFP, CDKL5^K33R-GFP, CDKL5^S162A,T164A,Y166A-GFP, or CDKL5^Y166F-GFP, and lf2 mutant cells expressing CDKL5-GFP were stained for flagella (acetylated tubulin, red) and CDKL5 (GFP, green top panels, gray in bottom panels). The cell bodies of all strains are green (or gray) due to autofluorescence. Wild-type CDKL5-GFP concentrates at the basal end of the flagellar shaft (arrows) as described by [2]. Note that all of the mutant CDKL5 proteins lack the major enrichment at the basal end of the flagellum but continue to show punctate stain along the flagellar shaft. Scale bar, 5 µm. Z projection of slices taken at 0.5-µm intervals. **(D)** Quantification of the CDKL5 pool at the base of the flagellar shaft. The quantification tools of ImageJ were used to measure fluorescence intensity in an approximately 1-µm circle at the base of each flagellum. $n > 100$ for each condition; one flagellum was measured per cell. ****$p \leq 0.0001$, *$p \leq 0.05$, as compared to control (#, top row), lf5 (#, second row), lf5 CDKL5^K33R-GFP (#, third row), or lf5 CDKL5^S162A,T164A,Y166A-GFP (# bottom row) by one-way ANOVA with Tukey's multiple comparisons post-hoc test. Violin plots show median (solid line) and quartiles (dashed lines). Underlying data can be found in S1 Data.

PLOS Biology

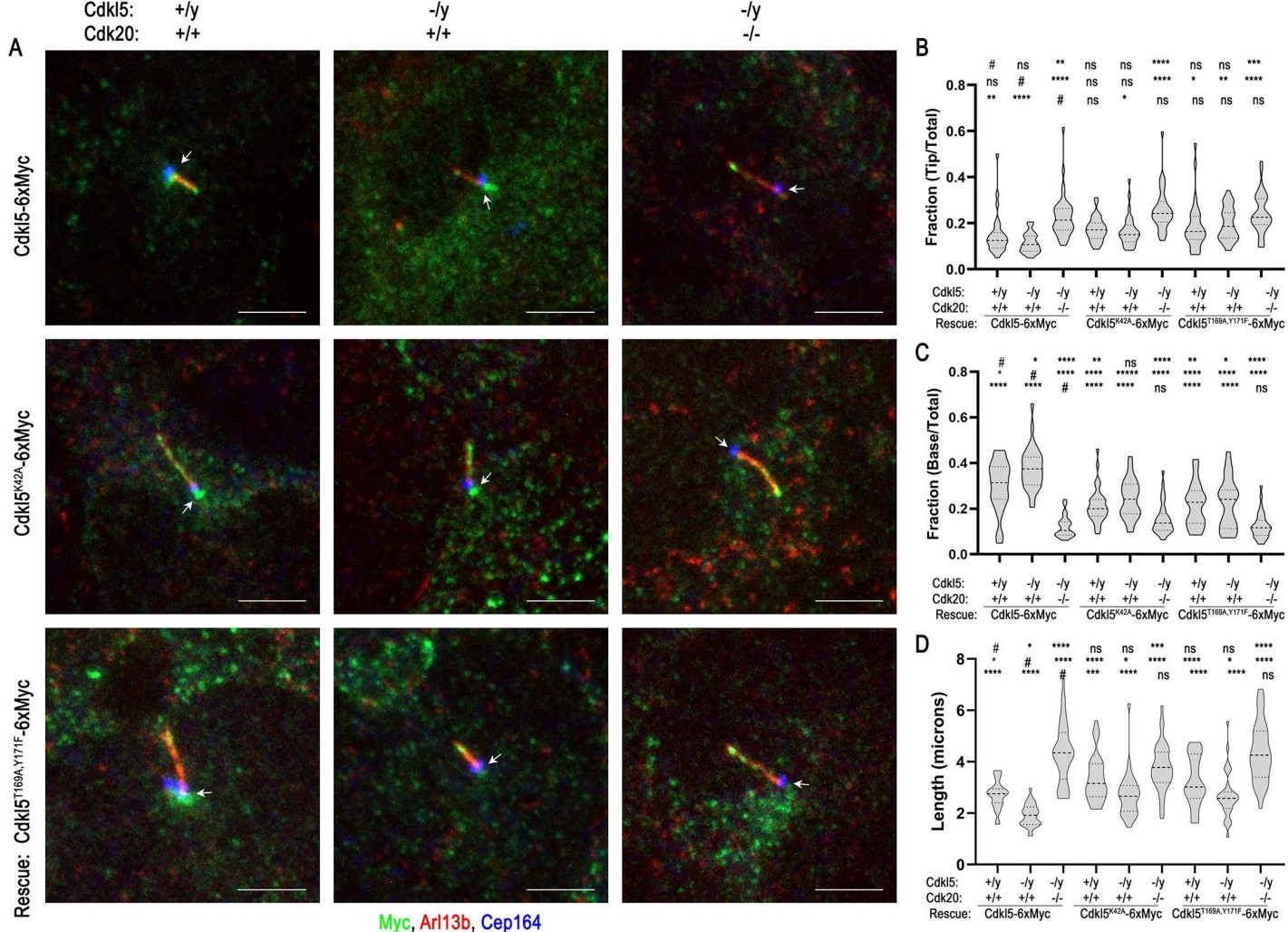

**Fig 8. Mammalian CDKL5 localizes in the basal body region outside the cilium but concentrates at the ciliary tip when Cdk20 is lost. (A)** Subciliary localization of mouse CDKL5-6xMyc, CDKL5$^{K42A}$-6xMyc, and CDKL5$^{T169A,Y171F}$-6xMyc in control, *Cdkl5* mutant, and *Cdkl5/Cdk20* double-mutant cells. Cells were transfected with the constructs, drug-selected, fixed, and stained for cilia (Arl13b, red), centrioles (Cep164, blue), and Cdkl5-6xMyc (Myc, green). Arrows mark the cilia bases. Scale bar, 5 μm. Z projection of slices taken at 0.37-μm intervals. **(B, C)** Quantification of the fraction of Cdkl5 signal located at the ciliary tip (B) or at the base (C) in the cells as exemplified in panel (A). To do this analysis, the measurement tools of FIJI were used to quantify the total signal in the cilium plus basal pool and the pools at the ciliary tip or base. $n > 25$ cilia for each condition. ****$p \le 0.0001$, **$p \le 0.01$, *$p \le 0.05$, ns: not significant as compared to control (#, top row), *cdkl5* mutant (#, middle row), or *Cdkl5/Cdk20* double-mutant cells (#, bottom row) by one-way ANOVA with Tukey's multiple comparisons post-hoc test. Violin plots show median (darker dashed line) and quartiles (dashed lines). Underlying data can be found in S1 Data. **(D)** Quantification of ciliary length in the cells as exemplified in panel (A). $n > 25$ cilia for each condition. ****$p \le 0.0001$, ***$p \le 0.001$, **$p \le 0.01$, *$p \le 0.05$, ns: not significant as compared to control (#, top row), *cdkl5* mutant (#, middle row), or *Cdkl5/Cdk20* double-mutant cells (#, bottom row) by one-way ANOVA with Tukey's multiple comparisons post-hoc test. Violin plots show median (darker dashed line) and quartiles (dashed lines). Underlying data can be found in S1 Data.

cells (Fig 7), CDKL5$^{K33R}$ was diminished at the flagellar base (orange arrowheads in Fig 9B) and enriched along the flagellar shaft and tip (blue arrowheads). Thus, CDKL5's kinase activity is required for the down-regulation of its transport once flagella reach steady-state. Steady-state IFT of CDKL5$^{Y166F}$ more closely resembled wild-type CDKL5 than CDKL5$^{K33R}$, although diffusion appeared more prominent (Fig 9C, 9F). Unlike CDKL5$^{K33R}$, a substantial pool of CDKL5$^{Y166F}$ accumulated at the base of the flagellar shaft, though in a more diffuse pattern than wild-type, and little or none localized to the tip

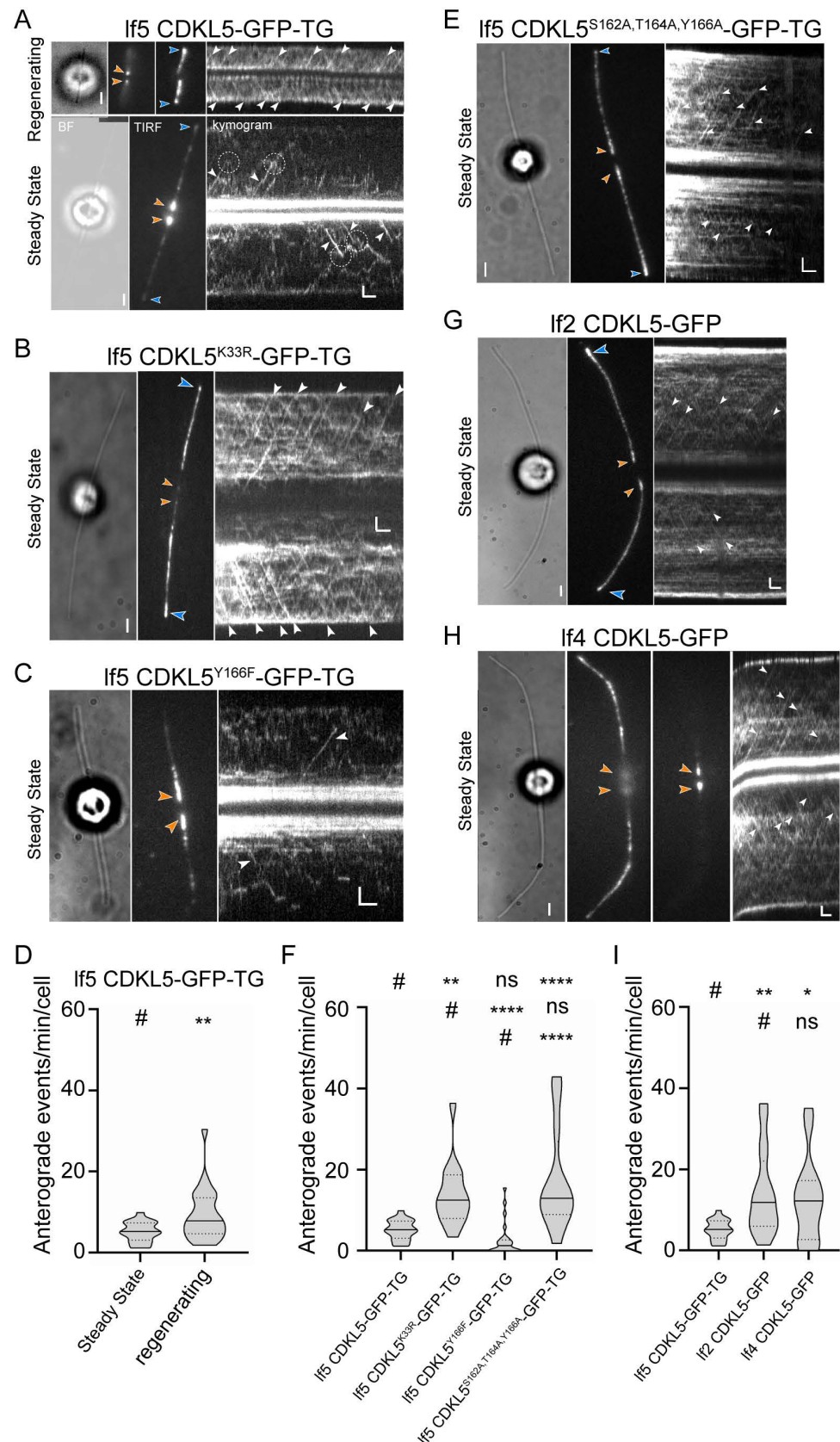

**Fig 9. CrCDKL5 is an IFT cargo in growing flagella but its transport is down-regulated in steady-state flagella. (A, B, C,** and **E)** Bright field, TIRF still images, and corresponding kymograms of CDKL5-GFP **(A)**, CDKL5$^{K33R}$-GFP **(B)**, CDKL5$^{Y166F}$-GFP **(C)**, and CDKL5$^{S162A,T164A,Y166A}$-GFP **(E)**, all expressed in the *lf5* mutant. The flagellar tips are marked with blue arrowheads, and orange arrowheads mark the flagellar bases. IFT events are marked with white arrowheads. The top row in A shows regenerating flagella; the TIRF images in this row show two focus levels, one in the plane of the flagellar tips and one in the plane of the flagellar bases. The bottom row in A shows steady-state flagella. The dashed circles in the kymogram of steady-state flagella mark events of CDKL5-GFP dissociating from IFT. Scale bar, 2 µm and 2 s. **(D)** Violin plot of the anterograde IFT frequencies of CDKL5-GFP in steady-state (*n* = 21) and regenerating (*n* = 52) flagella. **\*\****p* ≤ 0.01 as compared to steady-state flagella (#). Violin plots show median (solid line) and quartiles (dashed lines). Underlying data can be found in S1 Data. **(F)** Violin plot comparing the anterograde IFT frequencies of CDKL5-GFP, CDKL5$^{K33R}$-GFP, CDKL5$^{Y166F}$-GFP, and CDKL5$^{S162A,T164A,Y166A}$-GFP expressed as transgenes in the *lf5* mutant. (*n* = 21,14, 25, and 12, respectively). **\*\*\*\****p* ≤ 0.0001, **\*\*\****p* ≤ 0.001, **\*\****p* ≤ 0.01, ns: not significant as compared to control (#, top row), CDKL5$^{K33R}$ (#, second row), or CDKL5$^{Y166F}$ (#, bottom row). Violin plots show median (solid line) and quartiles (dashed lines). Underlying data can be found in S1 Data. **(G, H)** Bright field, TIRF still images, and corresponding kymograms of CDKL5-GFP in flagella of the *lf2* mutant (G) and the *lf4* mutant (H); two focus levels are shown for the latter. The arrowheads and bars are used as described in A–C. **(I)** Violin plot comparing the anterograde IFT frequencies of CDKL5-GFP in the lf5 CDKL5-GFP-TG, lf2 CDKL5-GFP, and lf4 CDKL5-GFP strains. (*n* = 21, 16, and 14, respectively). **\*\****p* ≤ 0.01, **\****p* ≤ 0.05, ns: not significant as compared to control (#, top row); ns, not significant compared to CDKL5-GFP in *lf2* mutant cells (#, bottom row). Violin plots show median (solid line) and quartiles (dashed lines). Underlying data can be found in S1 Data.

(Fig 9C). This combination of IFT-based transport and diffuse basal enrichment suggests that CDKL5$^{Y166F}$ retains partial kinase activity. Supporting this interpretation, CDKL5$^{S162A,T164A,Y166A}$ behaved more like CDKL5$^{K33R}$ than CDKL5$^{Y166F}$. Importantly, in steady-state flagella, CDKL5$^{S162A,T164A,Y166A}$ transport was strongly upregulated compared with wild-type and failed to undergo the tip-to-base redistribution characteristic of wild-type protein (Fig 9E, 9F). Thus, blocking CDKL5 kinase activity or mutating residues of the activation loop prevents the transition to the steady-state pattern of transport and localization observed with wild-type flagella.

To test whether CDKL5 localization and IFT depend on the length-regulatory kinases LF2 and LF4, we examined wild-type CDKL5-GFP in steady-state flagella of live *lf2* and *lf4* mutant cells. In the *lf2* background, the localization of CDKL5 resembled that of CDKL5$^{S162A,T164A,Y166A}$ in control cells, including diffuse enrichment at the flagellar base, substantial signal along the shaft, and strong accumulation at the tip (Fig 9G). In *lf2* cells, anterograde CDKL5 IFT was elevated compared to wild-type and matched the frequency seen with CDKL5$^{K33R}$ and CDKL5$^{S162A,T164A,Y166A}$ in control cells (Fig 9G, 9I). In the *lf4* background, CDKL5 showed frequent but low-processivity anterograde movements. The protein was concentrated at the flagellar base similar to wild-type but also found along the shaft and in a small pool at the tip (Fig 9H, 9I). Together, these results suggest that LF2 is more critical than LF4 for maintaining CDKL5 localization at the flagellar base, likely through regulation of its kinase activity. Nonetheless, both kinases contribute to the down-regulation of CDKL5 transport by IFT that normally occurs once flagella reach steady-state length.

## Global effects on flagellar composition and protein phosphorylation due to the lack of CDKL5

To understand how CDKL5 controls ciliary composition, we compared *lf5* flagella with wild-type flagella by tandem mass tag (TMT) labeling and MS. Analysis of two biological replicates (Experiments 8 and 9) revealed 55 ciliary proteins that were reproducibly increased and 43 ciliary proteins that were reproducibly decreased in *lf5* flagella (Fig 10 and S5 Data). Among the 55 increased proteins, 26 are IFT particle or motor proteins, indicating that abnormally active IFT may contribute to the long flagella phenotype. Of potential interest, PRMT1, a protein arginine methyltransferase implicated in IFT and flagellar growth [38], was elevated nearly 3-fold. Similarly elevated was METE1, which produces the methionine that is converted to the S-adenosyl methionine used by PRMT1 and other protein methyltransferases. METE1 was reported to be relatively low in steady-state flagella but elevated in regenerating flagella and highest in resorbing flagella [39]. Among the 43 decreased proteins, CDKL5 was barely detectable as expected in a *lf5* mutant. The length-regulatory kinase LF4 was also reduced. In addition, an assortment of motility proteins including dynein arm, radial spoke, and central apparatus subunits were reduced in the absence of CDKL5.

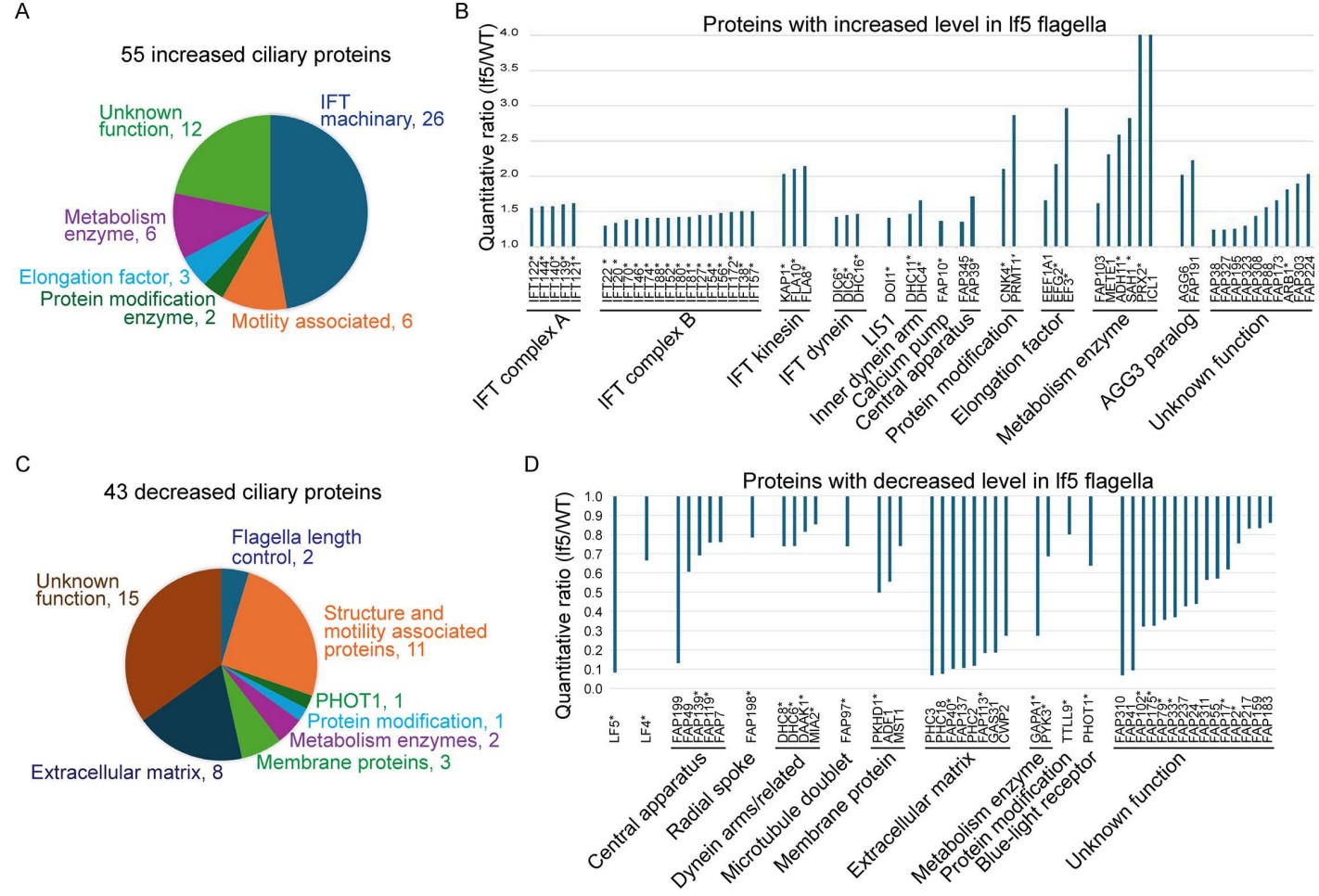

**Fig 10. CDKL5 has a global effect on ciliary composition.** Purified flagella from wild-type control (21gr) and lf5 cells were analyzed by MS. **(A, B)** Compared to the control, 55 ciliary proteins were increased in lf5 flagella. Two proteins, PRX2 and ICL1, had lf5/WT ratios of 6.66 and 24.81, respectively, well above the 4.0 maximum shown on the graph. Note that IFT particle and motor proteins were consistently increased in the absence of CDKL5. Asterisks in B indicate that these proteins have human homologs. Underlying data can be found in S5 Data on sheet "Increased." **(C, D)** Compared to the control, 43 ciliary proteins were decreased in lf5 flagella. Asterisks in D indicate that these proteins have human homologs. Underlying data can be found in S5 Data on the sheet "Decreased".

Since CDKL5 is a protein kinase, we compared the phospho-proteome of *lf5* and wild-type flagella and identified 43 sites in 36 proteins that have reduced phosphorylation in the absence of CDKL5 (Fig 11A and S6 Data). To determine a consensus sequence for CDKL5, we compared the amino acid composition at each of the positions to the overall abundance of that residue in the substrates as a whole. Residues R at −3, P at −2, and P at +1 are more than 8, 4, and 4 times as abundant in these positions as compared to the abundance in the whole protein set, although there was more permissivity at the +1 position, with hydrophobic residues preferred (Fig 11B, 11C). The resulting consensus R-P-x-[S/T]-P is very similar to the mammalian CDKL5 phosphorylation-consensus motif R-P-x-[S/T]-[A/G/P/S] [40]. To determine how well each of the candidate substrates matched the consensus, a numerical value was calculated based on the enrichment of the amino acid at each position (S6 Data, sheet "Motif Analysis"). The scores, while limited in number, showed a bimodal distribution with peaks at 15 and 22 and a valley at 19, suggesting that the 26 sites (in 23 proteins) with scores above 19 are likely direct targets of CDKL5 (Fig 11D). One of the proteins, the intraflagellar transport protein IFT74, has a

high-scoring consensus sequence centered on residue T34, which is in the highly basic N-terminal domain that forms part of the IFT74/IFT81 tubulin-binding domain required for efficient transport of tubulin into the cilium [41–43]. The reduced phosphorylation of IFT74's T34 in the absence of CDKL5 suggests a connection between CDKL5 and IFT of tubulin. This site is conserved between *Chlamydomonas* and humans.

Interestingly, CDKL5 itself has a CDKL5 consensus sequence centered on S584 near its C-terminus. The site was phosphorylated on CDKL5-GFP isolated from control cells but not on protein isolated from a *lf2* mutant, and not on kinase-dead CDKL5, indicating that phosphorylation at this site occurs by autophosphorylation after CDKL5 is activated LF2.

### FAP93 is a CDKL5-interacting protein

To identify proteins that interact directly with CDKL5, we immunoprecipitated CDKL5-GFP from whole-cell lysates of lf5 CDKL5-GFP-TG cells and subjected the precipitate to MS. Whole-cell lysates of untransformed *lf5* cells were treated identically as a control for nonspecific immunoprecipitation. Six proteins specifically co-precipitated with CDKL5 in two independent experiments (S7 Data). None of these proteins showed alterations in flagellar abundance in the absence of CDKL5 (S5 Data) and only FAP430 (Cre01.g009850.t1.2) had altered phosphorylation in the absence of CDKL5 (S6 Data). FAP430 is an uncharacterized coiled-coil protein that is not conserved in vertebrates. One of the co-precipitating proteins, FAP93, was recently shown to localize very similarly to CDKL5 [44]. HA-tagged FAP93 co-localized with CDKL5 near the proximal end of the flagellum (Fig 12A). Cells lacking FAP93 had a reduced amount of CDKL5 in the proximal flagellar pool and this could be rescued by the re-expression of GFP-FAP93-HA (Fig 12C–12E), indicating that FAP93 is required for anchoring CDKL5 at this site. Hwang and colleagues reported that *fap93* mutant cells have "near normal length" flagella, which we confirm (S7 Fig); thus, localization of CDKL5 at the base of the flagellum is not a prerequisite for flagellar length control. Whether FAP93 mutants have altered flagellar regeneration kinetics or other flagellar assembly defects is not known.

The four remaining proteins are mostly unstudied in the context of ciliary biology and only two (Cre02.g075900.t1.1 and Cre12.g499500.t1.1) are conserved in humans. These two are protein kinases similar to BRSK1 in humans and SNRK2 kinases in plants. BRSK1 is centrosome localized and phosphorylates gamma tubulin [45], whereas SNRK2 kinases are involved in stress responses and development [46]. Interestingly, four of the co-precipitated proteins—FAP430, FAP93, SNRK2C, and FAP365—as well as CrCDKL5, are among the 155 ciliary proteins reported to be methylated [47].

## Discussion

Virtually all neurons project a primary cilium from their soma [48], and disruption of this organelle is now recognized as a cause of diverse neurological disorders. These range from intellectual disability, seizures, and eating disorders to structural brain abnormalities including hydrocephalus, holoprosencephaly, and cerebellar hypoplasia. Increasing evidence is linking cilia to neuropsychiatric conditions, including Parkinson's disease [49], autism [50], Down syndrome [51], and bipolar disorder [52]. More recently, ciliary dysfunction has been implicated in neurodegenerative diseases such as ALS [53–55]. Work in both *Chlamydomonas* and mammalian systems strongly places CDKL5 within cilia, yet most studies of CDD have not considered how ciliary defects might contribute to disease. Here, we address this gap by examining how the loss of CDKL5 alters ciliary composition and function.

We chose to work on *Chlamydomonas* as a model for CDD not only because of its well-known advantages for studying cilia, but also because it clearly has a CDKL5 ortholog. Other model organisms like flies and worms have a single CDKL that groups with the CDKL1/2/3/4 branch of the family (Fig 1A). *Caenorhabditis* cdkl-1 localizes to the transition zone of neuronal cilia and controls ciliary structure and length [9,21,22]. In *Drosophila*, the loss of *Cdkl* causes seizures, hearing loss, climbing and flying defects, along with a shortened life span [23,24]. Cilia mutations in flies cause deafness and coordination defects [56], but whether the hearing loss and mobility defects in the *Cdkl* mutants are related to cilia remains to be determined. Few human variants have been found in the CDKL1/2/3/4 branch. However, patients

**A**

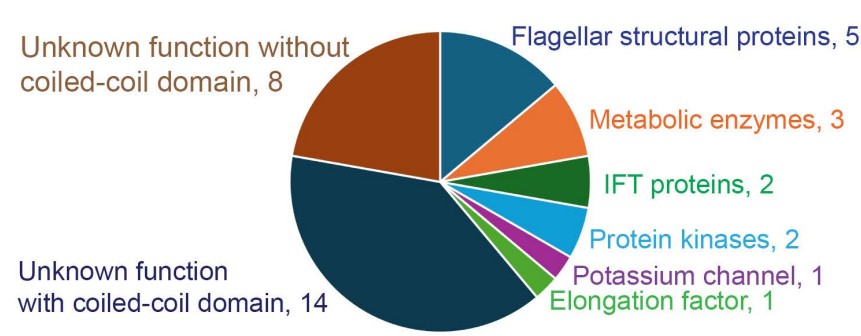

**B** **C**

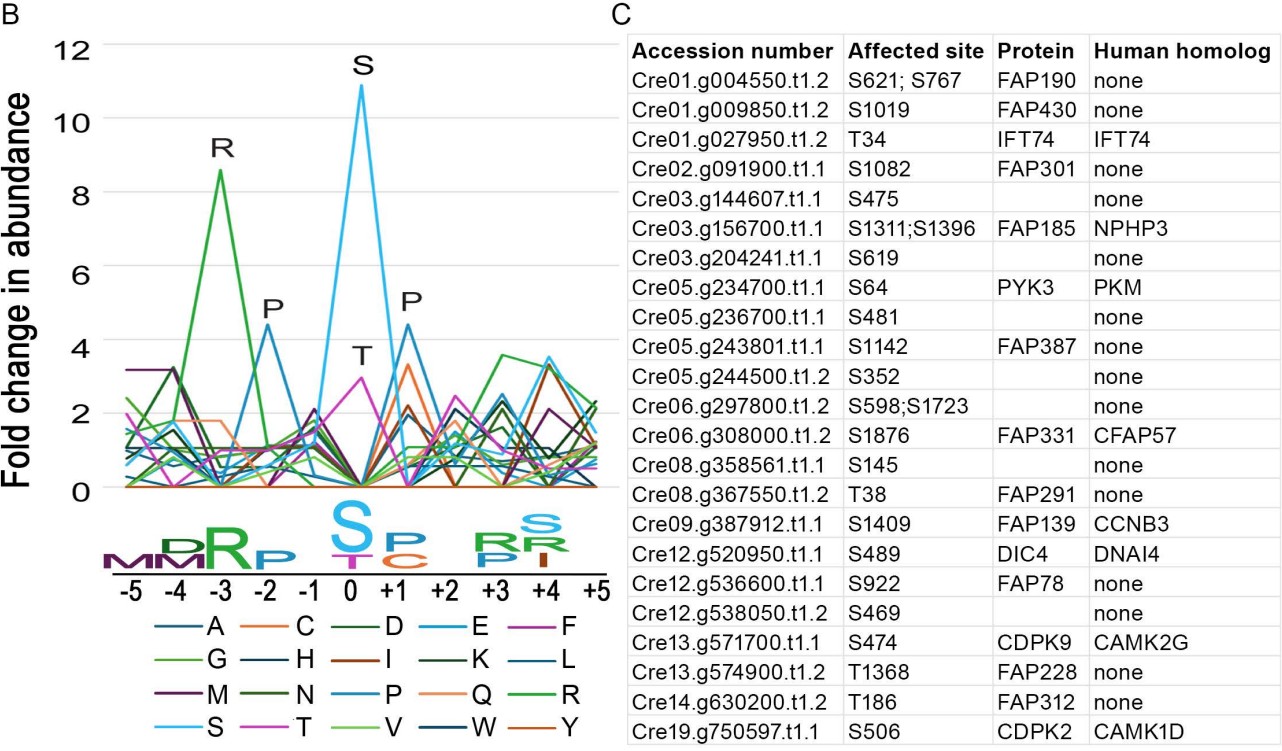

| Accession number | Affected site | Protein | Human homolog |
|---|---|---|---|
| Cre01.g004550.t1.2 | S621; S767 | FAP190 | none |
| Cre01.g009850.t1.2 | S1019 | FAP430 | none |
| Cre01.g027950.t1.2 | T34 | IFT74 | IFT74 |
| Cre02.g091900.t1.1 | S1082 | FAP301 | none |
| Cre03.g144607.t1.1 | S475 | | none |
| Cre03.g156700.t1.1 | S1311;S1396 | FAP185 | NPHP3 |
| Cre03.g204241.t1.1 | S619 | | none |
| Cre05.g234700.t1.1 | S64 | PYK3 | PKM |
| Cre05.g236700.t1.1 | S481 | | none |
| Cre05.g243801.t1.1 | S1142 | FAP387 | none |
| Cre05.g244500.t1.2 | S352 | | none |
| Cre06.g297800.t1.2 | S598;S1723 | | none |
| Cre06.g308000.t1.2 | S1876 | FAP331 | CFAP57 |
| Cre08.g358561.t1.1 | S145 | | none |
| Cre08.g367550.t1.2 | T38 | FAP291 | none |
| Cre09.g387912.t1.1 | S1409 | FAP139 | CCNB3 |
| Cre12.g520950.t1.1 | S489 | DIC4 | DNAI4 |
| Cre12.g536600.t1.1 | S922 | FAP78 | none |
| Cre12.g538050.t1.2 | S469 | | none |
| Cre13.g571700.t1.1 | S474 | CDPK9 | CAMK2G |
| Cre13.g574900.t1.2 | T1368 | FAP228 | none |
| Cre14.g630200.t1.2 | T186 | FAP312 | none |
| Cre19.g750597.t1.1 | S506 | CDPK2 | CAMK1D |

**Fig 11. Potential CDKL5 substrates and CDKL5 phosphorylation-consensus motif. (A)** Purified flagella from control (21gr) and lf5 cells were analyzed by MS to identify proteins with decreased phosphorylation when CDKL5 is missing. Forty-three sites in 36 proteins were found to have decreased phosphorylation. The proteins are grouped by function and predicted structural domains. See S6 Data for details. **(B)** To identify a CDKL5-phosphorylation-consensus motif, the 43 sites where phosphorylation levels were reduced in lf5 flagella were aligned with respect to the phosphorylation site (see sheet "Motif Analysis" in S6 Data). The X-axis represents position relative to the phosphorylation site (position 0). The Y-axis represents the fold change in abundance for each amino acid. To calculate the fold change of abundance for one amino acid, the percentage of this amino acid at one specific position among all the 43 sites is divided by the percentage of this specific amino acid among all the sequences from the 36 positive proteins. Peaks reveal the consensus sequence, which is R-P-X-S/T-P. Underlying data can be found in S6 Data on tab 'Motif Analysis'. **(C)** Potential CDKL5 substrates whose phosphorylation sites strongly match the consensus sequence (score >19). Human homolog data from ChlamyFP.org.

with *CDKL1* and CDKL2 variants have been reported that present with global developmental delay, intellectual disability, childhood-onset epilepsy, and dyspraxia—symptoms similar to those associated with CDD and other ciliopathies [23]. Whether these symptoms result from cilia dysfunction remain to be determined, but it seems likely from the observations

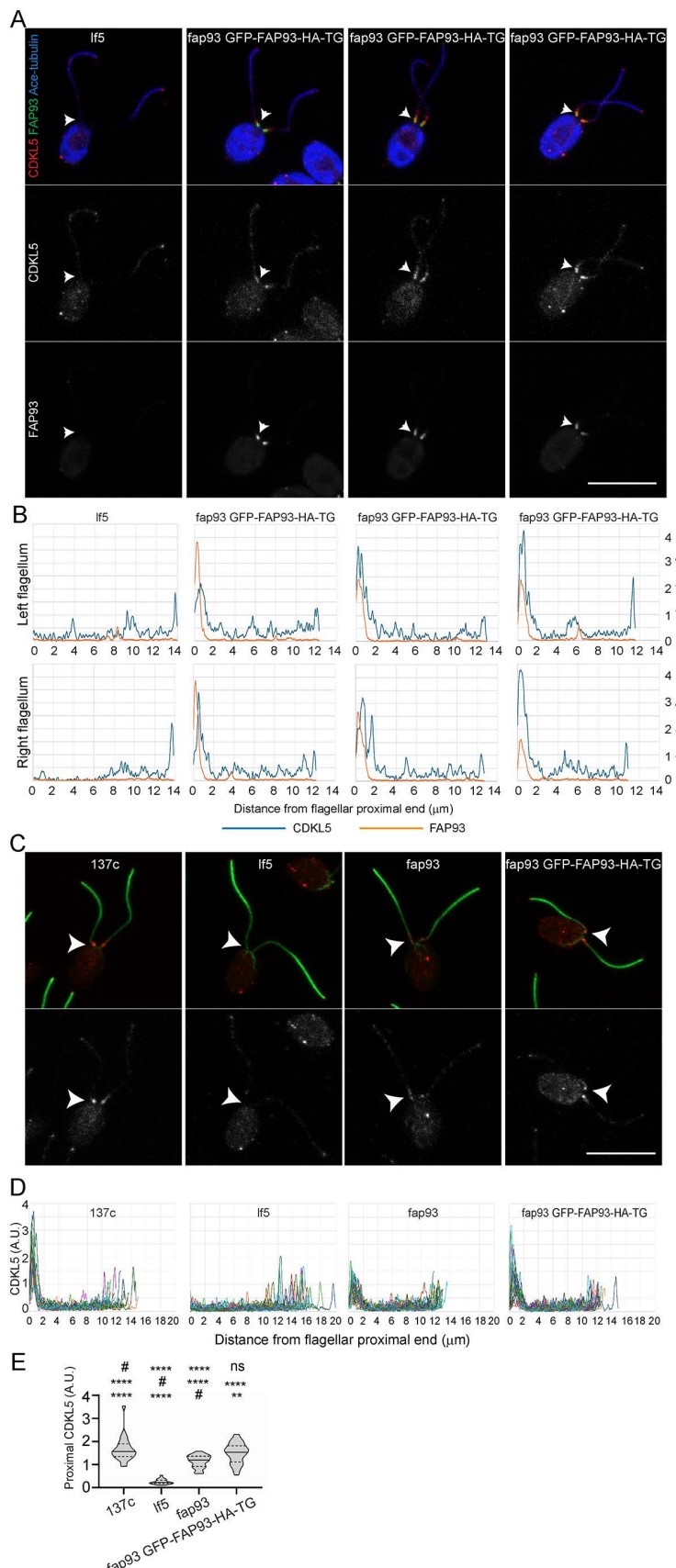

**Fig 12. FAP93 is required to concentrate CDKL5 at the proximal end of the flagellum. (A)** Cells of lf5 and fap93 GFP-FAP93-HA-TG strains were probed with anti-acetylated tubulin for flagella (blue in top row), anti-CDKL5 (red in top row, gray in second row), and anti-HA for FAP93 (green in top row, gray in third row). The cell bodies of all strains are blue and red (or gray) due to autofluorescence. CDKL5 and FAP93 concentrate at the proximal end of the flagella; arrowheads mark the proximal end of the left flagellum of each cell. Three examples of fap93 GFP-FAP93-HA-TG cells are shown. Scale bar, 10 μm. Z projection of slices taken at 0.5-μm intervals. **(B)** Panels below the images of each cell show the relative signal strength for CDKL5 (blue) and FAP93 (red) along the length of each flagellum. The quantification tools of ImageJ were used to draw a line along the length of the flagellum and measure the fluorescence intensity in 0.099-μm intervals. Underlying data can be found in S1 Data. **(C)** Control (137c), lf5, fap93, and fap93 GFP-FAP93-HA-TG cells were stained for flagella (acetylated tubulin, green) and CDKL5 (red in top panels, gray in bottom panels). The cell bodies of all strains are red (or gray) due to autofluorescence. Arrowheads mark the proximal end of the flagellar shaft where CDKL5 concentrates in control cells. Scale bar, 10 μm. Z projection of slices taken at 0.5-μm intervals. **(D)** Quantification of CDKL5 label along the flagella. The quantification tools of ImageJ were used to draw a line along the length of one flagellum per cell and measure the fluorescence intensity in 0.099-μm intervals. For each strain, the results for 25 flagella are superimposed. Underlying data can be found in S1 Data. **(E)** Quantification of the CDKL5 pool at the base of the flagellar shaft. The Y axis shows fluorescence intensity in an ~2-μm oval at the base of one flagellum from each of 25 cells of each strain. ****$p ≤ 0.0001$, **$p ≤ 0.01$, ns: not significant as compared to control (#, top row), lf5 (#, middle row), or fap93 (# bottom row) by one-way ANOVA with Tukey's multiple comparisons post-hoc test. Underlying data can be found in S1 Data.

in *Caenorhabditis* and *Drosophila*. Also, in support of this idea, the *Chlamydomonas* genes FLS1 and FLS2 in this clade are involved in ciliary disassembly [28,29]. Human CDKL5 can rescue the phenotypes of the *Drosophila* Cdkl mutant [23], but it is not known if there is any relationship between the cellular roles of CDKL5 and members of the CDKL1/2/3/4 clade in mammals.

## CDKL5 defects impair function of motile cilia

Human patients with CDD show an increased risk of respiratory disease [1], suggesting defects in motile cilia. Consistent with this observation, impaired ciliary motility was observed in mouse trachea and ependymal cilia and cilia from human nasal biopsies lacking CDKL5 [11]. Similarly, *Chlamydomonas CDKL5* mutants swim more slowly, likely due to less coordinated flagellar beating and altered waveform (Fig 2). Whether these defects arise from specific changes in ciliary components or are a general consequence of excessively long cilia remains unclear. Abnormally long cilia have been reported in a patient with primary ciliary dyskinesia, suggesting that cilia of increased length may function poorly [57]. Abnormal motility may also reflect intrinsic axonemal defects. Supporting this, MS revealed reduced flagellar abundance of several motility-related proteins when CDKL5 was absent (Fig 10).

## Regulation of CDKL5 via phosphorylation

CDKL5 is a CMGC-family protein kinase related to CDKs, MAPKs, GSKs, and CLKs. Members of this family typically contain TEY-motif activation loops, whose phosphorylation regulates kinase activity [30]. Studies of mammalian CDKL5 suggest that autophosphorylation of the TEY tyrosine is a key regulatory step. Mari *and colleagues* [18] showed that in vitro-translated HsCDKL5 could autophosphorylate, though the site was not identified. Lin *and colleagues* [17] further demonstrated autophosphorylation within the kinase domain on a tyrosine residue. Munoz and colleagues [16] provided stronger evidence, showing that wild-type HsCDKL5 was phosphorylated on a tyrosine, and that mutation of either the active site or the TEY tyrosine abolished this modification. Using a phospho-specific antibody against the TEY-motif, they confirmed that phosphorylation depended on both kinase activity and the presence of the TEY tyrosine. In *Chlamydomonas*, we observed that CDKL5 can also autophosphorylate in vitro (Fig 6) and that its activation loop is highly phosphorylated. However, unlike mammalian CDKL5, this phosphorylation persisted even when the catalytic site was inactivated, ruling out autophosphorylation at the activation loop (Fig 5).

Rather than autophosphorylation, our results indicate that the activation loop of *Chlamydomonas* CDKL5 is phosphorylated by the LF2 kinase (Fig 5). LF2 is similar to CDKL5 in that loss-of-function mutations cause abnormally long flagella [20]. Consistent with our finding that kinase-dead CDKL5 fails to concentrate at the basal flagellar end, loss of LF2 likewise causes CDKL5 to redistribute from the basal pool (Fig 7 and [2]). The mammalian homolog of LF2 is Cdk20

[37]. Mouse *Cdk20* mutants also display overly long cilia and show classic ciliopathy phenotypes, including defective Hedgehog signaling [37]. Cilia on these mutant cells accumulate IFT proteins and it has been suggested that Cdk20 works through CILK1, a related kinase that has been connected to cilia length control [37,58]. As in *Chlamydomonas*, mouse CDKL5 localizes to cilia but it is more enriched at the basal body. Nevertheless, loss of Cdk20 drives CDKL5 redistribution from the basal body region into the ciliary shaft indicating that LF2/Cdk20 has a conserved role in regulating CDKL5 localization.

Beyond the three phosphorylation sites in the activation loop, we identified 44 additional sites distributed across CDKL5, most of them located in the nonconserved C-terminal region. Intriguingly, residue S584 near the C-terminus was consistently phosphorylated in control cells but not in *lf2* mutants or in cells expressing kinase-dead CDKL5, suggesting that it may undergo autophosphorylation (S4 Data). If so, the C-terminal domain of CrCDKL5 may have an essential role in localizing the protein to the proximal part of the flagellum (see discussion of FAP93 below). In mammalian CDKL5, the C-terminal tail is phosphorylated at serine 407 [59] and serine 308 by Dyrk1a [60]. Mutation of S407 did not alter kinase activity but weakened CDKL5's interaction with PSD-95 at the synapse [59], while mutation of S308 changed the balance of CDKL5 between the cytoplasm and nucleus [60]. Additionally, deletion of the mammalian C-terminal region enhanced CDKL5 autophosphorylation, indicating that the tail has a negative regulatory role [17].

The finding that CDKL5 kinase activity controls its distribution raises the question of whether active CDKL5 modifies other proteins at the flagellar base to enable its binding or phosphorylates itself to promote association with local components. Our data support the latter, as kinase-dead CDKL5 expressed along with wild-type CDKL5 failed to localize to the flagellar base (S5 Fig). Phosphorylation of CDKL5 at S584 correlates with basal enrichment, making this site a plausible point of regulation. At the base, FAP93 is a strong candidate for the CDKL5 tethering component. FAP93 co-immunoprecipitates with CDKL5 (S7 Data) and, like CDKL5, is enriched at the base of the axoneme. Importantly, CDKL5 becomes dispersed in the absence of FAP93 (Fig 12 and [44]). The function of proximal CDKL5 localization remains uncertain. This pool lies just beyond the transition zone, so CDKL5 is lost when flagella are shed by abscission. We speculate that this loss could remove most CDKL5 from the cell, thereby eliminating a negative regulator and allowing faster regrowth. However, in mammals cilia do not appear to undergo active abscission, making a FAP93-dependent proximal enrichment of CDKL5 unnecessary.

## CDKL5 transport by IFT

We found that CDKL5 is transported by IFT during flagellar growth, but this transport is down-regulated once flagella reach steady-state (Fig 9). Although this down-regulation is likely important for controlling flagellar length, the underlying mechanism remains unclear. Down-regulation fails to occur in the absence of LF2 kinase, or with CDKL5 carrying active site or activation loop mutations, suggesting that the process depends on CDKL5's activity state. One possible explanation is that, in steady-state flagella, much of the cellular CDKL5 becomes sequestered at the proximal flagellum, thereby reducing the pool available for IFT. Consistent with this idea, proximal enrichment, like the down-regulation of transport, correlates with CDKL5 activity.

## Proteins downstream of CDKL5

To investigate how CDKL5 regulates ciliary length and function, we examined changes in ciliary protein composition and phosphorylation in its absence. MS identified 55 proteins enriched in flagella lacking CDKL5, about half of which are associated with IFT. This suggests that CDKL5 may directly regulate IFT, or alternatively, that IFT is upregulated to compensate for CDKL5 loss. Also, among the elevated proteins were PRMT1 and METE1, which are involved in protein methylation and localize along the cilium in punctate spots resembling the distribution of IFT particles. PRMT1 dynamically redistributes during ciliary growth and resorption, with enrichment at both the tip and near the base, mirroring CDKL5's

basal pool [38,39]. Since CDKL5 itself is methylated [47], these findings support a functional link between CDKL5 and the ciliary protein methylation machinery.

Conversely, 42 proteins were reduced in flagella lacking CDKL5. Interestingly, one of these was the length-regulatory kinase LF4. Previous studies have shown that LF4 is phosphorylated by LF2, but apparently not by CDKL5, and that LF4 is highly enriched in the flagella [61]. Taken together, these results suggest a complex interplay between CDKL5 and LF4 to regulate flagellar length.

We also identified a set of proteins with reduced phosphorylation in flagella lacking CDKL5. From these, we derived a CDKL5 consensus motif of R-P-x-[S/T]-P, which closely resembles the mammalian CDKL5 motif R-P-x-[S/T]-[A/G/P/S] [40]. Twenty-three proteins carried phosphorylation sites matching or approximating this consensus, making them likely to be direct CDKL5 substrates. Most are uncharacterized, but one of particular interest is the IFT-B subunit IFT74, which had reduced phosphorylation of T34 in its N-terminal domain. Together with IFT81, IFT74 forms a tubulin-binding module required for transporting tubulin into the flagellum [41,42]. IFT81 binds the globular domain of tubulin, while the highly basic N-terminus of IFT74 binds the acidic C-terminal tail of β-tubulin. Phosphorylation of T34 would decrease the N-terminal domain's positive charge and weaken tubulin-binding, suggesting a mechanism by which CDKL5 slows or halts tubulin transport as flagella approach steady-state. Mammalian IFT74 has a CDKL5 phosphorylation motif at the equivalent position suggesting that a similar mechanism may be at work to control cilia length in these organisms.

A similar regulatory mechanism is seen in *C. elegans*, where DYF-5 phosphorylates the N-terminal basic domain of IFT74, reducing its tubulin affinity. Mutation of the DYF-5 phosphosites to alanine produces abnormally long cilia, while phosphomimetic mutations shorten them [62]. The likely *Chlamydomonas* homolog of DYF-5 is MAPK7, a low-abundance, KCl-extractable ciliary protein [12,27]. MAPK7 is enriched in short, regenerating cilia, suggesting localization at the tip or base rather than uniform distribution along the shaft [63]. In mammals, the DYF-5 homologs CILK and MAK have been linked to IFT regulation and cilia length control [64–66].

Although CDKL5 is focused on the proximal end of the steady-state flagellum in *Chlamydomonas* and in the basal body region in mammalian cells, both foci are well positioned to facilitate phosphorylation of IFT74 as it is cycled through the organelle. In mammals, CDKL5's basal body distribution would ensure its close proximity to IFT74 during the assembly of anterograde IFT trains prior to their entry into the flagellum. In *Chlamydomonas*, activated CDKL5's concentration at the proximal end of the flagellum would ensure close proximity to IFT74 immediately before the IFT retrograde train is returned to the cell body. Therefore, both localizations may accomplish the same thing—to concentrate CDKL5 where it can efficiently engage IFT74 to control tubulin transport and ciliary length control.

## Summary

Our findings demonstrate that CDKL5 is a central regulator of ciliary function, linking its loss to ciliopathy-related phenotypes and providing new insight into CDD. We identified LF2 as the upstream kinase that activates CDKL5 via phosphorylation of its activation loop, with evidence for conservation through mammalian Cdk20, challenging the prevailing model that CDKL5 activation relies on autophosphorylation. Our proteomic and phosphoproteomic analyses combined with live-cell imaging showed CDKL5 is an important IFT regulator, and our finding that IFT74 phosphorylation is controlled by CDKL5 has important implications for tubulin transport and ciliary length control. Together, these results support the view that ciliary dysfunction should be considered an integral part of CDD pathology.

## Materials and methods

### Strains and media

*C. reinhardtii* strains utilized in this study are listed in Table 1. Cells were grown in Tris-acetate-phosphate (TAP) medium [67], M (minimal) medium I [68] modified to contain 0.0022 M $KH_2PO_4$ and 0.00171 M $K_2HPO_4$ [69], and M-N medium

(modified M medium lacking $NH_4NO_3$). Cells growing in liquid medium were aerated with a mixture of 5% $CO_2$ and 95% air. All cells were grown with a 14/10 h light/dark cycle.

The deletion producing the *lf5-2* allele was mapped by comparing PCR amplification of genomic DNA from wild-type (21gr, CC-1690) and *lf5-2* (CC-4560) with Gotaq (M300, Promega Corp, Madison, WI, USA) (S2 Data).

### *Chlamydomonas* plasmids

pLF5CsfGFPHyg was created by amplifying the 3′ end of the CDKL5 gene from pLF5CsfGFP [70] and inserting it into EcoRV-linearized plasmid pHyg3 [71]. Point mutations were created using PCR (600675, Herculase II Fusion DNA Polymerases, Agilent, Santa Clara, CA, USA) with primers in S1 Table to amplify fragments of pLF5CsfGFP, which were reassembled with NEBuilder HiFi DNA Assembly Master Mix (E2621, New England Biolabs, Ipswich, MA, USA). Reassembled plasmids were confirmed by restriction digest and Sanger sequencing. Details of plasmids used in this work are in S3 Table and SnapGene files will be provided upon request.

### GFP tagging of the endogenous *Chlamydomonas* *CDKL5* gene

TIM-tagging [70,72] was used to insert superfolder GFP at the C-terminus of the endogenous *CDKL5* gene. Donor DNAs for genome editing were created by PCR from pLF5CsfGFP [70] or pLF5CsfGFPHyg using primer pairs in S1 Table. Cells were transformed with donor DNA, Cas9 protein (1081058, Alt-R S.p. Cas9 Nuclease V3, IDT, Coralville, Iowa, USA), and LF5 sgRNA with guide sequence GGTGCTGTACCAGACCAATG [70] (A35534, TrueGuide Synthetic sgRNA, Invitrogen, Waltham, MA, USA) by electroporation.

### *Chlamydomonas* transformation

Cell transformation with either DNA or Cas9 RNP and donor DNA was carried out by electroporation [72]. Briefly, cells were pretreated with autolysin to remove cell walls and then incubated at 40°C for 30 min. Cells were washed once with TAP + 40 mM sucrose. The cells were then resuspended in TAP + 40 mM sucrose at a concentration of $2–7 \times 10^8$ cells/ml, and mixed either with 1.5 µg linearized plasmid in the case of DNA transformation or Cas9 RNP and donor DNA in the case of GFP tagging in a final volume of 125 µl. The mixture was electroporated in a 0.2-cm gap cuvette (1652086, Bio Rad Laboratories, Hercules, CA, USA) using an ECM 600 electroporator (BTX Harvard Biosciences, Holliston, MA, USA) at 350V, 25Ω, and 600 µF. The cuvette was incubated at 16°C for 1 h before the cells were rinsed out and incubated in 10 ml TAP + 40 mM sucrose under gentle rocking for 24 h in dim light. Cells were then concentrated and mixed with 0.5% agar in TAP media at 42°C and spread on TAP plates with either 10 µg/ml paromomycin (Paro) or 10 µg/ml Hygromycin (Hyg) (S3 Table). The transformants were first screened by PCR (S1 Table). Positive clones were further screened by western blotting using an antibody to GFP (11814460001, Roche, Basel, Switzerland).

### Protein sequence and structural analysis

Multiple protein sequences alignment and phylogenetic tree construction were done with Clustal software (www.genome.jp/tools-bin/clustalw) using only the conserved kinase domain for the alignment.

AlphaFold models were obtained from the AlphaFold server (www.alphafoldserver.com/) [73]. AlphaFold models were aligned with UCSF ChimeraX (www.rbvi.ucsf.edu/chimerax) [74,75].

### *Chlamydomonas* flagellar length, swimming speed, beat frequency, and waveform analysis

Flagellar length analysis was done by two methods. In the first method, synchronously grown cells were fixed in glutaraldehyde and imaged using phase optics as described [70]. For each strain, one of the two flagella was measured from 50 cells that clearly showed both flagella. A second method was then used to confirm the results for 137c, fap93, and fap93

GFP-FAP93-HA-TG flagellar length. In the second method, synchronously grown cells were collected by centrifugation, resuspended in HMS buffer (10 mM HEPES, pH 7.4, 5 mM $MgSO_4$, and 4% sucrose) and deflagellated with 10 mM dibucaine [69]. After deflagellation, EGTA and KCl were added to a final concentration of 0.5 and 25 mM, respectively. Flagella were separated from the cell bodies by centrifugation, applied to slides, and imaged using phase optics. The length of 50 flagella from each sample was measured using ImageJ [76].

For swimming speed and beat frequency analysis, cells were recorded under red-light illumination (2418, Gray Glass, Queens Village, NY, USA) to minimize phototactic responses. Imaging was performed using a Nikon Diaphot 200 inverted microscope equipped with a Nikon Fluor 10× Ph2 DL objective (NA 0.5). Videos were captured using a ASI174MM-COOL camera (ZWO Optical, Suzhou, China) controlled via SharpCap software (www.sharpcap.co.uk).

To measure swimming speed, video clips of swimming cells over a 1-s interval taken at 60 fps were analyzed using ImageJ. A minimum intensity Z projection was generated from each video clip to visualize the trajectories of swimming cells. For each strain, the distance along the trajectory of 50 individual cells was measured. Swimming speed was determined by dividing the trajectory length by the duration of the recording interval.

Beat frequency was determined by exploiting the characteristic motion of the cell body during the flagellar beat cycle: forward displacement occurs during the power stroke, followed by a slight backward movement during the recovery stroke. Each forward-backward displacement of the cell body corresponds to one complete flagellar beat. Video recordings of swimming cells were acquired at 500 fps over a 1-s interval, played back at 20 fps, and the number of beat cycles was manually counted for a minimum of 50 cells.

For waveform analysis, swimming cells were imaged by DIC microscopy at 1,000 fps. For analysis, we selected cells with both flagella in focus during several beat cycles. For each cell, the same flagellum was traced from each of 12 equally spaced frames covering one beat cycle and the traces then superimposed by aligning the cell body using Adobe Photoshop. Only flagella that remained entirely in focus throughout the beat cycle were selected for analysis. Two cells were analyzed for both the g1 wild-type strain and the lf5 CDKL5-GFP-TG rescued strain, while six cells were analyzed for the lf5 strain. To calculate the percentage of synchronized beats, each video recording of a swimming cell was analyzed to determine the total number of beats and the number of synchronized beats over the recording duration. For each cell line, 20 individual cells were examined.

The contrast of AVI videos was adjusted using ImageJ before conversion to MP4 videos using VLC 3.0.21 (www.video-lan.org).

## Western blotting

Whole-cell lysates were prepared by adding 5×-denaturing sample buffer (50 mM Tris, pH 8.0, 160 mM DTT, 5 mM EDTA, 50% sucrose, and 5% SDS) to concentrated cells in water to make the final buffer 1×. Flagella and cell bodies were separated and purified by the dibucaine method [69]. For CIP treatment, purified flagella samples were resuspended in HMDEK buffer (30 mM HEPES, pH 7.4, 5 mM $MgSO_4$, 1 mM DTT, 0.5 mM EGTA, and 25 mM KCl). Equal amounts of the same sample were either treated with or without Quick CIP (M0525S, New England Biolabs, Ipswich, MA, USA) in 1X rCutSmart buffer for 30 min at 37°C. The samples were heated at 80°C for 2 min to inactivate CIP. 5×-denaturing sample buffer was added to the samples to make the final buffer 1×. Whole-cell lysate and cell body samples were passed through a 26-gauge needle several times to shear genomic DNA. Samples were boiled for 5 min and then separated by regular SDS-PAGE or PhosTag SDS-PAGE using 50 μM phos-tag acrylamide AAL-107 (NC0232095, Fujifilm Irvine scientific, Santa Ana, CA, USA) in 6% acrylamide [77] and transferred to Immobilon P (EMD Millipore, Billerica, MA, USA) membranes. Membranes were probed with primary antibodies (S2 Table) for either 2 h at room temperature or overnight at 4°C, washed 4 times and probed with HRP-conjugated secondary antibodies (S2 Table) for 1 h at room temperature. Blots were developed using SuperSignal West Femto Maximus Sensitivity Substrate (34095, Thermo Fisher, Waltham, MA, USA) or SuperSignal West Dura Extended Duration Substrate (34075, Thermo Fisher, Waltham, MA, USA). Images were

taken using a FluorChem Q imager (Alpha Innotech, San Leandro, CA, USA) or a Universal Hood III imager (Bio Rad Laboratories, Hercules, CA, USA) and adjusted using Adobe Photoshop. All adjustments were applied to the whole image.

### *Chlamydomonas* immunofluorescence microscopy

Cells were fixed and stained in solution. Briefly, cells were fixed with 2.5% paraformaldehyde for 20 min at room temperature, treated with 0.05% NP-40 for 5 min, and then washed three times with 1× PBS buffer. The washed cells were then incubated in blocking buffer (5% [w/v] BSA [2960, Sigma–Aldrich, St. Louis, MO, USA], 1% [v/v] Fish Skin Gelatin [G7765, Sigma–Aldrich], 10% [v/v] goat serum [10000C, Gibco Thermo Fisher, Waltham, MA, USA]), in 1× PBS for 1 h at room temperature before treatment with primary antibody (S2 Table) in blocking buffer overnight at 4°C. The next day, cells were washed three times with blocking buffer. Treatment with secondary antibody (S2 Table) was done at room temperature for 2 h in the dark. Cells were then washed two times with blocking buffer and once with 1× PBS. Stained cells were applied to coverslips and mounted to slides with ProLong Gold Antifade Mountant with DNA Stain DAPI (P36941, Invitrogen, Carlsbad, CA, USA).

Fluorescent images were obtained with a Zeiss (Oberkochen, Germany) LSM900+ Airyscan confocal microscope with a 63× objective. If comparisons were to be made between images, the photos were taken with identical conditions and manipulated equally. For quantification of fluorescence intensity in cilia, z-stacks were flattened, and the region of interest was selected and quantified with the measure tools of ImageJ [78]. Local background was measured next to the region of interest and subtracted from the value.

### TIRF live-cell microscopy

TIRF microscopy was performed using a Nikon (Tokyo, Japan) Eclipse Ti-U microscope equipped with a 60X NA1.49 TIRF objective and through-the-objective illumination using a Triple Band LaserSet (ZET405/488/561m/x/rpc, Chroma Technology, Bellows Falls, VT, USA) excitation filter. The emission signal was cleaned up with a 520/35 nm BrightLine single-bandpass filter and a 560-nm edge BrightLine single-edge dichroic beamsplitter (Chroma Technology) using a W-VIEW Gemini (Hamamatsu, Iwata City, Japan) image-splitting device. Imaging was conducted at 10 fps using an iXON3 camera (Andor Technology, Belfast, Northern Ireland) controlled by NIS-Elements Advanced Research software (Nikon, Tokyo, Japan). Live imaging was performed in custom observation chambers. A 24 × 60 mm No. 1.5 coverslip was coated with a ring of petroleum jelly to create a chamber, into which 15 µl of cell suspension was introduced and allowed to settle for 0.5–3 min under bright light. The chamber was sealed by overlaying a 22 × 22 mm No. 1.5 coverslip with a 15-µl drop of imaging buffer (5 mM HEPES, pH 7.3, and supplemented with 3–5 mM EGTA). Imaging was performed at room temperature through the larger coverslip and videos were recorded using an iXon 887 EMCCD camera (Andor Technology, Belfast, UK) operated via the Nikon Elements imaging software. Image processing and analysis were conducted using imageJ. Kymograms were generated with KymoResliceWide plugin (imagej.net/KymoResliceWide). Analysis of anterograde IFT from kymograms was done by counting the number of IFT tracks in the flagella manually within a given time frame. Adobe Photoshop was utilized for contrast and brightness adjustments, and final figures were prepared using Adobe Illustrator.

### In vitro assay of CDKL5 autophosphorylation

CDKL5-GFP was prepared from flagella isolated by the dibucaine procedure as previously described [69]. Isolated flagella were resuspended in binding/dilution buffer (10 mM Tris/Cl pH 7.5, 0.5 mM EDTA, and 100 mM NaCl supplemented with protease inhibitor cocktail [1:100, P9599, Sigma-Aldrich, St. Louis, MO, USA] and phosphatase inhibitors [20 mM sodium pyrophosphate, 50 mM sodium fluoride]) and lysed by addition of 0.8% NP-40 and 0.5 M NaCl; the samples were incubated on ice for 30 min and mixed by vortexing and/or pipetting every 10 min. Chromotek magnetic

agarose beads (gtma, ProteinTech, Rosemont, IL, USA) (20–25 μl per flagella from 1–2 L of culture) were prepared by washing the beads three times in cold washing buffer (10 mM Tris/Cl pH 7.5, 150 mM NaCl, 0.05% NP-40, and 0.5 mM EDTA); proteinase inhibitors were added to the beads during the last wash. The flagellar lysate was diluted with binding buffer to adjust the NP-40 concentration to 0.3% and NaCl to 200 mM, added to the beads, and incubated with rotation at 4°C overnight. Then, the beads were sedimented by low-speed centrifugation and repeatedly (5×) washed with wash buffer supplemented with proteinase inhibitors. For some preparations, flagella were flash frozen in liquid nitrogen and stored at −80°C until use.

To assay for autophosphorylation, the following reagents were added sequentially to 20 μl of beads with attached CDKL5-GFP: 30 μl of 2× kinase buffer (50 mM Tris HCl pH 7.5, 16 mM MOPS, 145 mM NaCl, 10 mM MgCl$_2$, and 2 mM EGTA, 5% glycerol) with fresh phosphatase inhibitors (0.8 mM sodium pyrophosphate, 10 mM sodium fluoride), protease inhibitor cocktail 1:100, 0.4 mM DTT, and 0.08 mM ATP-γS (ab138911, Abcam, Cambridge, UK). The mixture was incubated for 1 h at 30°C. The reactions were terminated by adding 2.5 mM p-nitrobenzyl mesylate (ab138910, Abcam, Cambridge, UK) and incubated at room temperature for 1 h. The samples were resuspended with 5× sample buffer and heated at 85°C for 10 min. After separation by SDS-PAGE and western blotting, the membranes were stained with anti-thiophosphate ester antibody (ab92570, Abcam, Cambridge, UK) to detect the thiophosphorylation and anti-GFP (G10362, Thermo Fisher, Waltham, MA, USA) to detect CDKL5-GFP.

## Mass spectrometry analysis

A total of 11 MS experiments (S4 Table) were carried out to analyze CDKL5 phosphorylation states, flagellar proteome and phospho-proteome in the absence of CDKL5, and CDKL5-interacting proteins. For the analysis of CDKL5 phosphorylation states, purified wild-type or mutant CDKL5 from whole-cell lysates of lf5 CDKL5-GFP-TG, lf5 CDKL5$^{K33R}$-GFP-TG, or lf2 CDKL5-GFP cells were used. For the determination of the flagellar proteome and phospho-proteome in the absence of CDKL5, purified flagellar samples from 21gr and lf5 cells were compared using a TMT approach. To identify CDKL5-interacting proteins, the products of anti-GFP immunoprecipitations from whole-cell lysates of lf5 CDKL5-GFP-TG and lf5 cells were compared. Details on biological sample preparation, CIP treatment, protein digestion, LC-MS/MS, TMT-LC-MS/MS, and downstream data analysis for the 11 MS experiments are described in S1 Text.

Note that the results for CDKL5-GFP from lf5 CDKL5-GFP-TG cells, serving as wild-type controls in Experiments 3 and 4 and untreated controls in Experiments 5 and 6, were included in the nonquantitative overview of CDKL5 phosphorylation sites described in the subsection "CDKL5 is extensively phosphorylated, including at three sites within the activation loop" and in S3 Data.

## Mice

Mouse line B6.129(FVB)-Cdkl5tm1.1Joez/J [31] (021967, Jackson Laboratory, Bar Harbor, ME, USA) was maintained as hemizygous males or heterozygous females by recurrent mating to wild-type C57Bl/6J (000664, Jackson Laboratory, Bar Harbor, ME, USA).

Mice were genotyped with WT-F34149 (GGAAGAAATGCCAAATGGAG plus WT-R34150 (GGAGACCTGAAGAG-CAAAGG) to yield a 248-bp product from the wild-type alleles and MT-F34151 (CCCTCTCAGTAAGGCAGCAG) plus MT-R34152 (TGGTTTTGAGGTGGTTCACA) to yield a 347-bp product from the mutant allele.

## Study approval

Mouse research was carried out at UMass Chan Medical School with IACUC approval (PROTO201900265). This IACUC follows the regulations of the U.S. Department of Agriculture Animal Welfare Act and the standards/principles of the Public

Health Service Policy on Humane Care and Use of Laboratory Animals, AVMA Guidelines on Euthanasia, U.S. Government Principles for the Utilization and Care of Vertebrate Animals Used in Testing, Research and Training, and the Guide for the Care and Use of Laboratory Animals.

**Mammalian cell culture and immunofluorescence microscopy**

Mutant MEFs were isolated from E13.5 mutant embryos and immortalized with the large T antigen from SV40 virus. Cell lines were genotyped as described for the mouse and were additionally sexed with primers SryF (TTGTCTAGAGAGCATGGAGGGCCATGTCAA) plus SryR (CCACTCCTCTGTGACACTTTAGCCCTCCGA), which gives a 273-bp product in male cells and no product in female cells. Lines 26548.4T +/y and 26548.5T −/y were used for experiments (S2 Fig). MEFs and their derivatives were grown at 37°C in 5% $CO_2$ in Dulbecco's modified Eagle's medium (11995-065, DMEM; Gibco Thermo Fisher, Waltham, MA, USA) with 5% FBS (F0926, Sigma-Aldrich, St. Louis, MO, USA) and 1% Penicillin-Streptomycin (15140-122, Gibco Thermo Fisher, Waltham, MA, USA). All DNA constructs were transfected into the cells via lentiviral infection followed by drug selection to create stable cell lines.

Motile ciliated cells were obtained by differentiating ependymal cells from mouse cortex [79]. To do this, the cortex from newborn pups was minced, digested in 0.25% Trypsin-EDTA (25200-056, Gibco Thermo Fisher, Waltham, MA, USA) plus 0.1% Type IV DNAse (D5025, Sigma-Aldrich, St. Louis, MO, USA) for 10 min. After digestion, tissue was dispersed by pipetting through a 1-ml pipet, centrifuged at 600 rcf for 2 min, and the pellet resuspended in DMEM containing 20% FBS and 1% Penicillin-Streptomycin. The cells were plated on glass coverslips and cultured for 3 days. At that time, the medium was replaced with DMEM without serum and the cells were cultured until motile cilia could be observed, which typically took 7–10 days.

Cells for immunofluorescence microscopy were grown on acid-washed glass coverslips. The cells were fixed for 15 min in 2% paraformaldehyde (15710, EM Sciences, Hatfield, PA, USA), 0.05 M Pipes, 0.025 M Hepes, 0.01 M EGTA, and 0.01 M $MgCl_2$ (pH 7.2) followed by a 2-min extraction with 0.1% Triton X-100 in the same solution. For some antibodies, an antigen-retrieval step of 0.05% SDS in PBS for 5 min was included at this point. After two brief washes in TBST (0.01 M Tris, pH 7.5, 0.166 M NaCl, and 0.05% Tween 20), the cells were blocked with 1% bovine serum albumin (2960, Sigma-Aldrich, St. Louis, MO, USA) in TBST for 1 h and then incubated with the primary antibodies either overnight at 4°C or for 2 h at room temperature. The cells were then washed 4 times with 1% BSA/TBST over ~30 min. Next, the cells were incubated with 1:2000 dilutions of Alexa fluor-conjugated secondary antibodies (Invitrogen, Waltham, MA, USA) (S2 Table) for 1 h and washed 4 times with 1% BSA/TBST over ~30 min followed by a brief wash with TBST. The cells were then mounted with ProLong Gold Antifade (P36930, Molecular Probes, Eugene, OR, USA) and visualized by fluorescence microscopy. Images were taken on a Zeiss LSM900 Airyscan confocal (Oberkochen, Germany) with 63× or 40× objectives. If comparisons were to be made between images, the photos were taken with identical conditions and manipulated equally. For quantification of fluorescence intensity in cilia, z-stacks were flattened, and the region of interest was selected and quantified with the measure tools of ImageJ [78]. Local background was measured next to the region of interest and subtracted from the value.

Cells were confirmed to be of mouse origin and monitored for mycoplasma contamination by PCR [80] and DAPI staining.

**Mammalian cloning**

Plasmids were assembled by T5 exonuclease-dependent assembly (a.k.a. TEDA) [81] into the pHAGE lentiviral backbone [82]. Mutations were generated by PCR amplification with mutated primers and the products assembled as above. All inserts were fully sequenced and matched NCBI reference sequences or expected mutant forms. Plasmids are listed in S3 Table and SnapGene files will be provided upon request.

## Lentivirus production

Lentiviral-packaged pHAGE-derived plasmids [82] were used for transfection. These vectors were packaged by a third-generation system comprising four distinct packaging vectors (Tat, Rev, Gag/Pol, VSV-g) using HEK 293T cells as the host. DNA (Backbone: 5 µg; Tat: 0.5 µg; Rev: 0.5 µg; Gag/Pol: 0.5 µg; VSV-g: 1 µg) was delivered to the HEK cells using calcium phosphate precipitates. After 48 h, supernatant was harvested, filtered through a 0.45-µm filter, and added to ~50% confluent cells. After 24 h, cells were selected with blasticidin (Bsd, 60 µg/ml), puromycin (Puro, 1 µg/ml), or nourseothricin (Nat, 50 µg/ml).

## Mammalian genome editing

Guide RNAs were selected from the Brie library [83] or designed using CHOPCHOP [84]. Corresponding oligonucleotides were cloned into lentiCRISPR v2 Puro (Addgene plasmid #52961, deposited by Feng Zhang; [85]) or lentiCRISPR v2 PuroP93S (BL245) and screened by sequencing. The vector lentiCRISPR v2 PuroP93S is like its parent except for a pro-line to serine mutation in the puromycin N-acetyl-transferase gene, which increases its resistance to puromycin (https://www.addgene.org). The vectors were packaged into lentiviral particles and transfected into MEF cells. After selection, individual cells were sorted into 96-well plates by flow cytometry. Mutant clones were identified by immunoblotting. Rescue experiments used a CDKL5-expression construct containing silent mutations to obscure recognition by the guide.

## Supporting information

**S1 Fig. Alignment of kinase domains of *Chlamydomonas* and human CDKL5.** Box marks the activation loop. Red K is the active site lysine and red Y is the activation loop tyrosine.
(TIF)

**S2 Fig. Specificity of D-12 Santa Cruz monoclonal antibody. (A, B)** Western blots of MEF extracts probed with CDKL5 Santa Cruz monoclonal antibody clone D-12 (A) and γ-tubulin antibodies (B). Arrow marks the expected band. *marks a nonspecific band. 11479.6T is the control line used for generation of CRISPR knockouts. 26548.1-.6T are lines derived from embryos of the B6.129(FVB)-Cdkl5tm1.1Joez/J mouse. 26548.4T (+/y) (Control) and 26548.5T (−/y) (Mutant) were used for further work. **(C)** Immunofluorescence of wild-type and *Cdkl5*-mutant fibroblasts labeled with CDKL5 Santa Cruz monoclonal antibody clone D-12 (green in left panel, gray in right panel), Arl13b (red in left panel), and γ-tubulin (pink in left panel). Scale bar is 5 microns.
(TIF)

**S3 Fig. Failure to rescue beat frequency is not caused by a failure to localize to flagella or by the GFP tag. (A)** Western blots of flagella samples from wild-type (21gr), *lf5*, lf5 CDKL5-GFP-TG, three different transformants of lf5 CDKL5$^{K33R}$-GFP-TG, and three different transformants of lf5 CDKL5$^{Y166F}$-GFP-TG cells were probed with anti-CDKL5 or anti-GFP to reveal CDKL5 proteins. The same set of samples was diluted 1:25 and probed with anti-α-tubulin as a loading control. **(B)** Beat frequency of wild-type cells (g1) and three independent lines (CDKL5-GFP) created by CRISPR in the g1 background. Fifty cells were measured for each cell line. ns: not significant as compared to wild-type (#) by one-way ANOVA with Tukey's multiple comparisons post-hoc test. Violin plots show median (solid line) and quartiles (dashed lines). Underlying data can be found in S1 Data.
(TIF)

**S4 Fig. Fibroblast CRISPR knockouts of *Cdkl5* have long cilia. (A)** Mouse embryonic fibroblast line 11479.6T was transfected with lentiviral guide constructs BL848, BL849, and BL850. After drug selection, the cells were single-cell sorted into 96 well dishes and then screened by western blotting with a Cdkl5 antibody (Santa Cruz D-12) using a γ-tubulin antibody as loading control. Asterisks (*) mark cell lines that were subsequently analyzed. **(B)** Control and the

knockout cells were stained for cilia (acetylated α-tubulin and γ-tubulin in red, Arl13b in green). Scale bar, 5 μm. Z projection of slices taken at 0.37-μm intervals. **(C)** Quantification of cilia length in the cells described in B. $n > 100$ for each condition. ****$p \leq 0.0001$ as compared to control (#) by one-way ANOVA with Tukey's multiple comparisons post-hoc test. Violin plots show median (solid line) and quartiles (dashed lines). Underlying data can be found in S1 Data. **(D)** Rescue. Quantification of cilia length in the wild-type, BL850c3, and BL850c3 rescued with wild-type CDKL5 mutated to resist the guide RNA (MS236). $n > 100$ for each condition. ***$p \leq 0.001$, ns: not significant as compared to control (#) by one-way ANOVA with Tukey's multiple comparisons post-hoc test. Violin plots show median (solid line) and quartiles (dashed lines). Underlying data can be found in S1 Data.
(TIF)

**S5 Fig. CDKL5$^{K33R}$-GFP distribution is not affected by wild-type CDKL5. (A)** Wild-type (g1) cells untransformed or expressing CDKL5-GFP or CDKL5$^{K33R}$-GFP were stained for flagella (acetylated tubulin, red) and CDKL5 (GFP, green in top panels, gray in bottom panels). The cell bodies of all strains are green (or gray) due to autofluorescence. Wild-type CDKL5-GFP concentrates at the basal end of the flagellar shaft (arrows) while CDKL5$^{K33R}$-GFP does not, similar to what was observed when these constructs were expressed in *lf5* mutant cells (Fig 7). **(B)** Quantification of the CDKL5 pool at the base of the flagellar shaft. The quantification tools of ImageJ were used to measure fluorescence intensity in an approximately 2-μm oval at the base of each flagellum. $n > 100$ for each condition; one flagellum was measured per cell. ****$p < 0.0001$ as compared to control (#, top row) or g1 CDKL5-GFP-TG (#, bottom row) cells by one-way ANOVA with Tukey's multiple comparisons post-hoc test. Violin plots show median (solid line) and quartiles (dashed lines). Underlying data can be found in S1 Data.
(TIF)

**S6 Fig. Knockout of *Cdk20* from *Cdkl5* mutant fibroblasts. (A)** Mouse embryonic fibroblast line 26548.5T was transfected with lentiviral CRISPR guide constructs MS216, MS217, and MS218. After drug selection, the cells were single-cell sorted into 96-well dishes and then screened by western blotting with a Cdk20 antibody using an actin antibody as loading control. **(B)** Diagram of mouse Cdk20 showing the position of the kinase domain (red), active site lysine (K), positions of the three guides (MS216, MS217, and MS218) and the location of the antigen used to generate the antibody used.
(TIF)

**S7 Fig. Flagellar length in *fap93* is normal.** Flagella lengths of wild-type cells (137c), *fap93* cells, and fap93 GFP-FAP93-HA-TG cells are shown. One flagellum from each of 50 cells was measured for each cell line. No significant difference (ns) was identified among the three strains by one-way ANOVA with Tukey's multiple comparisons post-hoc test. Top row shows the pairwise comparison to wild-type cells (#) and bottom row shows the pairwise comparison to *fap93* cells (#). Violin plots show median (solid line) and quartiles (dashed lines). Underlying data can be found in S1 Data.
(TIF)

**S1 Data. Underlying data.** Individual values underlying the graphs shown in the figures along with the corresponding statistical analysis.
(XLSX)

**S2 Data. Analysis of the *lf5-2* locus.** Schematic of *Chlamydomonas* chromosome 12 surrounding the *CDKL5* locus. Scale bar is 1 kb. Positions of the PCR products are shown in black if the region was not deleted in *lf5-2* and fuchsia if it was deleted. Genomic DNA from wild-type (21gr, CC-1690) and lf5-2 (CC-4560) was amplified with GoTaq (M7122, Promega Corp, Madison, WI, USA) using the primers listed below. In each gel, lane 1 is wild-type, lane 2 is *lf5-2* and lane 3 is a no DNA added control. Ladder is 1 kb extended DNA ladder (N3239, NEB, Ipswich, MA, USA). In addition to CDKL5, Cre12.g538250, and Cre12.g538200 were also deleted. Cre12.g538250 is defined at JGI as sucrose 6-glucosyltransferase and is only conserved within *Chlamydomonadales*. Cre12.g538200 is defined at JGI as 50S

ribosome-binding GTPase (MMR_HSR1)//Serine hydrolase (FSH1) (FSH1)//RWD domain (RWD)//Obg-like GTPase YGR210-like, G4 motif-containing domain (YGR210-like_G4) and is conserved in algae and invertebrates. Neither gene has been connected to flagella in any study.
(DOCX)

**S3 Data.** *Chlamydomonas* **CDKL5 phosphorylation sites. Protein Sequences:** this sheet contains protein sequences for wild-type CrCDKL5, CrCDKL5-GFP, CrCDKL5[K33R]-GFP, CrCDKL5[Y166F]- GFP, and CrCDKL5[S162A,T164A,Y166A]-GFP. **CrC-DKL5 Phosphorylation Sites:** this sheet contains lists of candidate phosphorylation sites and confirmed phosphorylation sites identified in six different mass spectrometry analyses ("MS Experiments 1–6") of CrCDKL5 purified from whole-cell lysates of strain lf5 CDKL5-GFP-TG. Column "All candidate sites" lists all the S, T, and Y residues of CrCDKL5. Column "Confirmed" lists phosphorylated sites identified in at least one analysis. Column "Number of Experiments Identified in" lists the number of experiments in which the site was identified as phosphorylated. MS Experiments 1–6 are biological repeats. For MS Experiment 5, there were three technical repeats. Column "MS Experiment 5 combined" lists the phosphorylated sites identified in one or more of the technical repeats, which are shown as columns "MS Experiment 5_1", "MS Experiment 5_2", and "MS Experiment 5_3". For MS Experiment 6, there were 4 technical repeats. Column "MS Experiment 6 combined" lists the phosphorylated sites identified in one or more of the technical repeats, which are shown as columns "MS Experiment 6_1", "MS Experiment 6_2", "MS Experiment 6_3", and "MS Experiment 6_4". An empty cell indicates that the phosphoisoform was not detected in that experiment. 100% probability was used as a cutoff for positive peptides. * indicates that one of the two candidate phosphorylation sites in the peptide was phosphorylated, but which one could not be determined. **Peptide 155–170:** this sheet lists all the phosphoisoforms found for peptide 155–170 of CrCDKL5 from whole-cell lysates of strain lf5 CDKL5-GFP-TG. In column "Peptide 155-170 phosphoisoform," a lower-case letter indicates that the residue is phosphorylated. Specific isoforms identified in each experiment are indicated in columns "MS Experiment 1" through "MS Experiment 6." See sheet "CrCDKL5 phosphorylation sites" for a detailed description of experiments and columns. An empty cell indicates that the phosphoisoform was not detected in that experiment. 100% probability was used as a cutoff for positive peptides.
(XLSX)

**S4 Data.** *Chlamydomonas* **CDKL5 phosphorylation in lf5 CDKL5-GFP-TG (control), lf2 CDKL5-GFP, and lf5 CDKL5[K33R]-GFP cells.** S4 Data shows the datasets for MS experiments 3, 4, 6, and 7. **All CDKL5 peptide isoforms**: a lists all the peptide isoforms (both phosphorylated and nonphosphorylated) identified in two sets of biological samples (MS experiments 3 and 4) consisting of purified CrCDKL5 from strains lf5 CDKL5-GFP-TG, lf2 CDKL5-GFP, and lf5 CDKL5[K33R]-GFP-TG. Column "Start position" lists the first amino acid of each peptide isoform. Column "Stop position" lists the last amino acid of each peptide isoform. Column "Phosphoisoform" shows the phosphorylated residues of each phosphopeptide; "Non" indicates peptide was not phosphorylated. Columns D–I show precursor intensities of each peptide isoform in each sample; numbers were normalized based on total peptide intensities from CDKL5. "NA" indicates that the peptide isoform was not detected. Value of 0 ("0.00E + 00") indicates that the peptide isoform was identified but no quantitative data acquired due to low-abundance. Isoforms from the activation loop peptide (aa 155–170) are highlighted in yellow; the nonphosphorylated isoform of this peptide was consistently detected at much higher abundance in samples from the lf2 CDKL5-GFP strain than from the other two strains. Highlighted in green are phosphorylated sites detected only in samples from the lf5 CDKL5-GFP-TG strain, suggesting that these sites are auto-phosphorylated. Of these, residues surrounding only S584 match the CDKL5 phosphorylation-consensus sequence (see Fig 11). Peptides beyond amino acid 622 are from GFP. **All CDKL5 peptides**: a list of the sum of all the isoforms (both phosphorylated and nonphosphorylated) for the same peptide from MS Experiment 3 and 4. Value of 0 ("0.00E + 00") indicates that the peptide was either not identified or no quantitative data acquired due to low-abundance. An empty indicates that the value for the denominator to calculate the ratio is zero. Peptide 155–170 is highlighted in yellow. **CDKL5 (WT and K33R) CIP peptides**: a list of the sum of all

the isoforms (both phosphorylated and nonphosphorylated) for the same peptide from MS Experiment 6 and 7. Value of 0 ("0.00E + 00") indicates that the peptide was either not identified or no quantitative data acquired due to low-abundance. An empty indicates that the value for the denominator to calculate the ratio is zero. Peptide 155–170 is highlighted in yellow.
(XLSX)

**S5 Data. Proteins with altered abundance in *lf5* mutant flagella. MS Experiment 8 All Proteins:** a list of proteins identified in MS Experiment 8. Empty cells mean no quantitative data acquired. **MS Experiment 9 All Proteins:** a list of proteins identified in MS Experiment 9. Empty cells mean no quantitative data acquired. **All Proteins Identified**: a list of proteins identified in MS Experiments 8 and 9 (S4 Table). Empty cells mean no quantitative data acquired. **Increased:** only increased proteins are listed. The proteins with "High" confidence (https://chlamyfp.org/ChlamyFPv2/cr_read_sql.php) are shown in Fig 10. **Decreased:** only reduced proteins are listed. The proteins with "High" confidence (https://chlamyfp.org/ChlamyFPv2/cr_read_sql.php) are shown in Fig 10.
(XLSX)

**S6 Data. Proteins with altered phosphorylation in *lf5* mutant flagella. MS Experiment 8_PhosphoPeptides:** a list of all the phosphorylated peptides identified in MS Experiment 8. **MS Experiment 9_PhosphoPeptides:** a list of all the phosphorylated peptides identified in MS Experiment 9. **All Positives:** a list of phosphorylation sites that are identified as decreased in both experiments. The cutoff criterium for adjusted abundance ratio of lf5 to −21gr is 1. Among these sites, the sites that have both ratios smaller than 0.549 are colored light blue, between 0.549 and 0.667 are colored orange, between 0.667 and 0.833 are colored gray. See Materials and methods for a detailed explanation. **Positives 2 Fold:** a list of phosphorylation sites that are identified as decreased in both experiments. The cutoff criterium for adjusted abundance ratio of lf5 to −21gr is 0.549. See Materials and methods for a detailed explanation. Sites with a score above 19 are shown in Fig 11C. **Motif Analysis:** a list of the sequences surrounding the phosphosites of the proteins included in Positives 2 Fold and the calculation of the motif score.
(XLSX)

**S7 Data. Proteins that co-immunoprecipitate with CrCDKL5. All Proteins:** a list of all proteins identified in three technical repeats of each of two immunoprecipitation experiments (MS Experiment 10 and MS Experiment 11) with each protein's total precursor intensity reported for each repeat. "#N/A" means that the protein was not identified in that experiment. Calculation of ratio and *P* value is described in the Materials and methods section. "Yes" in "Positive" columns means that the protein met the following criteria: the ratio of total precursor intensity in the experimental sample (lf5 CDKL5-GFP-TG) to that in the control sample (lf5) was larger than 10, and the *P* value was smaller than 0.01. CrCDKL5-GFP was the bait protein. Human keratin proteins and catalase were included in the sequence database to monitor contamination and to improve the accuracy and reliability of the MS protein identifications. **Positive Proteins:** a list of proteins that were positive in both immunoprecipitation experiments with their total precursor intensities reported for each technical repeat. In addition to the bait proteins (CrCDKL5-GFP and GFP), 6 proteins were identified. Interestingly, four of them as well as CrCDKL5 are among the 155 ciliary proteins reported to be methylated by King and colleagues [47]. PMID: 38696262. The proteins are sorted according to their lf5 CDKL5-GFP-TG to lf5 ratios in Experiment 10.
(XLSX)

**S1 Raw Images. Uncropped blots used to create** Figs 4, 6, 7, **S2**, **S3**, **S4**, **and** **S6**.
(PDF)

**S1 Table. *Chlamydomonas* primers.** Primers used for amplification of *Chlamydomonas* sequences.
(DOCX)

**S2 Table. Antibodies.** Antibodies used in this work.
(DOCX)

**S3 Table. Plasmids.** Plasmids created in this study.
(DOCX)

**S4 Table. MS experiments.** Overview of all of the MS runs analyzed in this study.
(DOCX)

**S1 Text. MS Methods.** Detailed methods for MS analysis.
(DOCX)

**S1 Video. Video clip of wild-type cells (21 gr) swimming.** The videos are time-stamped, and the resolution is 0.615 µm/pixel.
(MP4)

**S2 Video. Video clip of lf5 cells swimming.** The videos are time-stamped, and the resolution is 0.615 µm/pixel.
(MP4)

**S3 Video. High-speed video clip of a wild-type cell (g1) swimming to illustrate waveforms.**
(MP4)

**S4 Video. High-speed video clip of a rescued cell (lf5 CDKL5GFP-TG) swimming to illustrate waveforms.**
(MP4)

**S5 Video. High-speed video clip of an lf5 cell swimming to illustrate waveforms.**
(MP4)

**S6 Video. High-speed video clip of an lf5 cell swimming to illustrate waveforms.**
(MP4)

**S7 Video. High-speed video clip of an lf5 cell swimming to illustrate waveforms.**
(MP4)

**S8 Video. High-speed video clip of an lf5 cell swimming to illustrate waveforms.**
(MP4)

## Acknowledgments

We thank Dr. Paul A. Lefebvre for generously supplying antibodies to CrCDKL5 as well as the plasmid pHA-LF2. We thank Dr. Stephen J. Eyles from the Mass Spectrometry Core Facility at the University of Massachusetts Amherst and Dr. Laurie Parker from the Department of Biochemistry Molecular Biology and Biophysics at the University of Minnesota for their advice on MS. We are grateful to the Sanderson Center for Optical Experimentation at the UMass Chan Medical School for use of their optical resources. The Vermont Biomedical Research Network Proteomics Facility (RRID: SCR_018667) is supported through NIH grant P20GM103449 from the INBRE Program of the National Institute of General Medical Sciences. Flow Cytometry Resources at UMass Chan were supported by National Institutes of Health NIH S10OD028576. AlphaFold 3 model alignment was performed with UCSF ChimeraX, developed by the Resource for Biocomputing, Visualization, and Informatics at the University of California, San Francisco, with support from National Institutes of Health R01-GM129325 and the Office of Cyber Infrastructure and Computational Biology, National Institute of Allergy and Infectious Diseases. Chlamydomonas strains for this research were obtained from the Chlamydomonas Resource Center, funded by the US National Science Foundation.

# Author contributions

**Conceptualization:** Yuqing Hou, Oranti Ahmed Omi, Michael W. Stuck, Bryan A. Ballif, Karl F. Lechtreck, George B. Witman, Gregory J. Pazour.

**Data curation:** Yuqing Hou, Oranti Ahmed Omi, Michael W. Stuck.

**Formal analysis:** Yuqing Hou, Oranti Ahmed Omi, Michael W. Stuck, Ying-Wai Lam, Anna M. Schmoker, Son N. Nguyen, Maria Paz Gonzalez-Perez, Bryan A. Ballif, Karl F. Lechtreck, George B. Witman, Gregory J. Pazour.

**Funding acquisition:** Karl F. Lechtreck, George B. Witman, Gregory J. Pazour.

**Investigation:** Yuqing Hou, Oranti Ahmed Omi, Michael W. Stuck, Xi Cheng, Bethany Walker, Ying-Wai Lam, Anna M. Schmoker, Son N. Nguyen, Maria Paz Gonzalez-Perez, Bryan A. Ballif, Karl F. Lechtreck, George B. Witman, Gregory J. Pazour.

**Visualization:** Yuqing Hou, Oranti Ahmed Omi, Michael W. Stuck.

**Writing – original draft:** Yuqing Hou, Oranti Ahmed Omi, Karl F. Lechtreck, George B. Witman, Gregory J. Pazour.

**Writing – review & editing:** Yuqing Hou, Oranti Ahmed Omi, Michael W. Stuck, Xi Cheng, Bethany Walker, Ying-Wai Lam, Anna M. Schmoker, Son N. Nguyen, Maria Paz Gonzalez-Perez, Bryan A. Ballif, Karl F. Lechtreck, George B. Witman, Gregory J. Pazour.

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
