## [Editor Report · Decision Letter 0]

26 Jun 2025

Dear Greg,

Thank you for submitting your manuscript entitled "LF2 phosphorylation of the CDKL5 activation loop controls CDKL5’s activity" for consideration as a Research Article by PLOS Biology.

Your manuscript has now been evaluated by the PLOS Biology editorial staff as well as by an academic editor with relevant expertise and I am writing to let you know that we would like to send your submission out for external peer review.

Once your full submission is complete, your paper will undergo a series of checks in preparation for peer review. After your manuscript has passed the checks it will be sent out for review. To provide the metadata for your submission, please Login to Editorial Manager (https://www.editorialmanager.com/pbiology) within two working days, i.e. by Jun 30 2025 11:59PM.

Kind regards,

Ines

--

Ines Alvarez-Garcia, PhD

Senior Editor

PLOS Biology

---

## [Decision Letter · Decision Letter 1]

22 Aug 2025

Dear Dr Pazour,

Thank you for your continued patience while your manuscript "LF2 phosphorylation of the CDKL5 activation loop controls CDKL5’s activity" was peer-reviewed at PLOS Biology. Please accept my sincere apologies for the delays that you have experienced during the peer review process. Please note that I am currently handling your submission since my colleague Ines Alvarez-Garcia is currently away from the office this week. Your manuscript has now been evaluated by the PLOS Biology editors, an Academic Editor with relevant expertise, and by three independent reviewers.

In light of the reviews, which you will find at the end of this email, we would like to invite you to revise the work to thoroughly address the reviewers' reports.

As you will see, Reviewer’s #2 and #3 are generally positive about the manuscript but ask that additional textual and reporting edits are provided regarding data interpretation and discussing alternative possibilities to explain the finding. On the other hand, Reviewer #1 is more critical about the clarity and presentation/structure of the manuscript, its overall novelty and the lack of controls to characterize the genetic models. After discussions with the academic editor, we agree that the clarity and structure of the manuscript should be strengthened during revision and that the requested controls are included.

Given the extent of revision needed, we cannot make a decision about publication until we have seen the revised manuscript and your response to the reviewers' comments. Your revised manuscript is likely to be sent for further evaluation by all or a subset of the reviewers.

**IMPORTANT - SUBMITTING YOUR REVISION**

*Re-submission Checklist*

*Published Peer Review*

*PLOS Data Policy*

*Blot and Gel Data Policy*

Best regards,

Richard

Richard Hodge, PhD

rhodge@plos.org

On behalf of:

Ines Alvarez-Garcia, PhD

REVIEWS:

Reviewer #1: CDKL5 is part of a family of evolutionarily ancient kinases distantly related to cyclin-dependent kinases. CDKL5 in particular has attracted considerable attention due to its association with Cdkl5 Deficiency Disorder (CDD), a neurodevelopmental disorder characterized by epileptic seizures with an incidence of ~1:40,000, rare but considerably more common than eg Bardet-Biedl Syndrome (BBS, 1:140,000), a classical ciliopathy that is moreover genetically heterogeneous, with mutations in >20 loci. CDKL5 was originally linked to cilia and ciliary length control in a study in Chlamydomonas from 2013 (Tam et al., Mol Biol Cell). Further work by Mustafa Sahin and Cecilia Lo's labs in 2022 (Di Nardo et al., Neurosci Res; Faubel et al., Acta Neuropathol) showed that motile cilia in both CDD patients and knockout mice were elongated and poorly motile, with impaired cilia-driven flows in the mouse brain inducing seizure-like activity. Despite these clear data linking cellular function and clinical presentation and further indications of a role in ciliary length control for this family of kinases from studies on the paralog CDKL1 in C. elegans (Park et al., Curr Biol 2021), only 7 of the 776 largely clinical studies on CDKL5 to be found on PubMed make any mention of cilia in their title or abstract, highlighting a disconnect between applied research and basic cell biology largely conducted in model organisms. Further work on this kinase with a view to better understand its ciliary function is therefore clearly welcome.

The present study by the labs of Karl Lechtreck, George Witman and Gregory Pazour represents a follow-up on the work by Tam et al., further characterizing CDKL5/LF5 and its link to another kinase also involved in ciliary length control, CDK20/LF2 (Tam et al., J Cell Biol 2007), both in Chlamydomonas and to a lesser extent also in mice. The authors report that CDKL5 localization and function in ciliary length control in Chlamydomonas requires its kinase activity and that LF2 phosphorylates CDKL5's activation loop to regulate its kinase activity. Vertebrate CDK20 reportedly similarly regulates CDKL5 localization to the ciliary base in mice. Finally, the authors perform differential ciliary proteomics from wild-type and CDKL5 mutant Chlamydomonas and CDKL5 immunoprecipitations, identifying IFT proteins as increased in abundance in loss of function mutants and FAP93 as a potential regulator of CDKL5 localization to the ciliary base. Most intriguingly, the authors report CDKL5 phosphorylation of a residue within the IFT74 tubulin-binding domain, providing a potential molecular mechanism by which the kinase could control ciliary length.

Overall, then, there is plenty of data within the manuscript's sprawling 12 figures and 5 supplemental figures and experiments on the whole are conducted to a reasonable standard. My main criticism is that the manuscript feels like a grab bag of disconnected experiments, written in a way that is not immediately accessible to the reader. There is no obvious storyline. If published in its present format this would not serve its stated cause to raise the profile of CDD as a potential ciliopathy.

Main points

1. Clarity. Throughout the manuscript the authors use unnecessarily complicated language, making it extremely difficult to follow their arguments. Experiments are also presented in a highly idiosyncratic manner. Just to highlight a few examples:

p5/6

The results would appear to start with examining the phylogenetic distribution of CDKL kinases (p5) or at the latest when examining CDKL5 localization in motile and non-motile cilia (p6), yet this paragraph and the entirety of Fig 1 is first discussed in the Introduction (although the authors return to this theme on p8). Why?

p8/9

"Mammalian CDKL5 localizes to cilia and defects in it cause abnormally long cilia" (p8).."These data indicate that CDKL5's role in cilia length control is conserved in mammalian primary cilia." (p9)

The only novel aspect here is that the kinase dead mutant does not rescue, yet this aspect is not highlighted.

p12

For the most part, the phosphorylation patterns of CDKL5-GFP expressed in a lf2 background (i.e. lf2 CDKL5-GFP cells) and CDKL5K33R expressed in a lf5 background (i.e. lf5 CDKL5K33R-GFP-TG cells) were similar to that for CDKL5-GFP expressed in a lf5 background (i.e. lf5 CDKL5-GFP-TG cells)

Can this not be expressed in simpler terms?

p14

"Ability of CDKL5 to autophosphorylate in vitro is dependent upon in vivo expression in the presence of LF2 but not LF4"

What is this title supposed to mean?

p17

"To test the involvement of Cdk20 in regulating Cdkl5 localization, we removed Cdk20 from Cdkl5 mutant fibroblasts and rescued with the Cdkl5-6xMyc, Cdkl5K42R-6xMyc, and Cdkl52A-6xMyc constructs."

It is a little strange to refer to the expression of an unrelated GFP transgene as a 'rescue'.

p17

"To determine if mammalian Cdkl5's ciliary localization is regulated like Chlamydomonas CDKL5, we mutated the active site (Cdkl5K42R) and mutated the two phosphosites in the activation loop (Cdkl52A)."

The 2A mutant already appeared on p9, where it was not properly introduced ("Overexpression of the wild-type gene reduced the cilia length to slightly less than normal whereas expression of kinase-dead Cdkl5 did not alter cilia length but expression of Cdkl5 with mutated activation-loop phosphorylation sites was able to rescue. (Fig 3C)").

p19

"In support of this, the pattern for CDKL53A-GFP was much closer to that of CDKL5K33R-GFP than to CDKL5Y166F-GFP: compared to full-length lf5 CDKL5-GFP-TG flagella, CDKL53A-GFP transport by IFT was strongly upregulated, there was much less accumulation of the protein at the flagellar base, and there was some accumulation at the flagellar tip (Fig 9E, F)."

It is almost impossible to parse this sentence.

p21

"The site was phosphorylated in CDKL5-GFP from whole-cell lysates of lf5 CDKL5-GFP-TG cells but not from lf2 CDKL5-GFP cells, and not in CDKL5K33R from lf5 CDKL5K33R -GFP-TG cells, indicating that phosphorylation at this site occurs by autophosphorylation following activation of CDKL5 by LF2 (S2 Table and see section "Phosphorylation at the CrCDKL5 activation loop is controlled by LF2 and does not depend on autophosphorylation")."

Again, much more complicated than necessary.

Fig S1B

Why label a blot 11479.6T and 26548 1T-6T rather than Control and KO lines 1-6?

Fig S1C

Why show panel C, functionality of GFP CRISPR Chlamydomonas lines, in the same figure as panel B, characterizing CDKL5 mutant MEFs?

Fig 3B/C

There is no need to label an image panel with a plasmid number (eg MS234) which is nowhere referred to in the main text. Also -/y might be technically correct, but could simply be described as - with the hemizygous nature clarified in the legend. Finally, labeling one transgene K42A, the other 2A is confusing when the latter refers to T169A,Y171A.

Fig S2B/C

Why label knockout MEF cell lines and rescue condition in a way that is not accessible to the reader?

Fig 4A

This schematic fails to show phosphosites or any other feature of the protein (eg kinase domain) in a way that would be accessible to the reader.

Fig 5 (and S3)

This appears a sub-optimal way to present mass spec data. If the intent is to allow the reader to compare the levels of specific (phospho-)peptides in different experimental conditions, this is not easily done from either the tables or dot plots.

Fig 10D

Why invert the bar graphs?

Fig 11B

Potential substrate recognition motifs are not usually shown in this manner. Sequence logos would be much more interpretable.

2. Mechanistic insight/Novelty. In their abstract and concluding paragraph, the authors highlight their findings of CDKL5 localization to cilia and defects in ciliary length and motility in CDKL5 mutants and conclude that their results provide "..strong evidence that CDKL5 dysfunction contributes to ciliopathy-related phenotypes, supporting the hypothesis that CDD is a ciliopathy." (p33) As mentioned above, CDKL5 localization and ciliary length and motility phenotype has been previously reported, both in Chlamydomonas and in mice. Furthermore, there is quite convincing data supporting impaired ciliary motility in the brain as the underlying cause of the epileptic seizures in CDD patients, which goes far beyond what is shown in this manuscript. What is at least to some extent novel in their manuscript is the following:

1) CDKL5 kinase activity is shown to be required for its function - although this is perhaps unsurprising, given that this has been previously shown to be the case for CDKL-1 in C. elegans (Canning et al., Cell Rep 2018).

2) LF2 phosphorylates CDKL5 within the activation loop and thereby regulates its kinase activity - this is perhaps the most substantive finding in this manuscript, although again perhaps unsurprising, given that LF2 was shown by Tam et al., 2013 to affect CDKL5 localization in Chlamydomonas and CDK20 knockouts display elongated cilia with an accumulation of IFT proteins at their tips in RPE1 cells (Noguchi et al., PLOS One 2021 - incidentally a paper not cited by the authors).

3) Active CDKL5 in Chlamydomonas localizes to the ciliary base of mature cilia, whereas inactive kinase is unmoored and accumulates along cilia and at their tips, as it normally does in regenerating cilia - this had previously been shown for the wild-type protein by Tam et al. in their 2013 study; that the inactive kinase remains delocalized is interesting; however, the functional significance of this observation is unclear since it is perhaps not conserved in vertebrates (Fig 8) and delocalization in FAP93 mutants does not impact flagellar length (Fig S5).

4) Differential proteomics and IP interactomics identifies potential CDK5L targets/regulators - although these are either not further characterized or as in the case of FAP93's tethering function apparently dispensable for CDKL5 function. Regarding the ciliary accumulation of IFT proteins in CDKL5 mutants, something similar was shown for CDK20 mutants by Noguchi et al., where this was interpreted as impaired retrograde IFT, so again not entirely unprecedented.

While CDKL5 is undoubtedly a clinically relevant protein, the added mechanistic insight here in my view is limited. What to me would strengthen the manuscript considerably is a better understanding of the relationship between CDKL1 and CDKL5, which seem to perform the same function but are not redundant. Are the phenotypes additive? Similarly, is CDK20 merely an upstream regulator of CDKL5 (and 1?)? If so, there should again not be an additive phenotype in double mutants. These were actually generated by the authors (Fig 8), but their ciliary lengths were not assessed. Is retrograde IFT impaired? The authors have the tools to address these questions, which would actually advance our understanding of the function of this kinase family.

Other points

3. Characterizing the lf5 mutant

The authors describe the lf5 mutant as a null mutant (eg p7). This mutant was originally generated by Tam et al., 2013 (there referred to as DKD6 or lf5-2). In that puplication the authors failed to detect any transcript, but also did not map the genomic alteration. If there is residual expression of a truncated version of the protein or the mutation takes out more than just CDKL5, this would throw the failure to fully rescue the mutant using a GFP transgene (eg Fig. 2C, D) into a different light. Have the authors considered either sequencing the lf5-2 mutant or generating a clean gene deletion?

4. Characterizing the vertebrate MEF knockouts

While the authors do show blots to confirm full-length CDKL5 is no longer expressed in their MEF cell line knockouts (Fig S2) I could not find any such blot or IF images for CDK20. There is also no information on which exons were targeted for either CDKL5 or CDK20. They do report the sequences of the guides in Table S8, but one would expect at least a schematic showing where these are in the gene, also relative to the epitopes of the commercial antibodies the authors used to detect the two proteins.

5. CDKL5 localization in mice

p6 "Similar to what was observed in Chlamydomonas, mouse Cdkl5 localizes to both non-motile primary cilia and to motile cilia, although we did not observe the prominent enrichment at the proximal end of mammalian cilia that is observed in Chlamydomonas (Fig 1C, D)." This finding, also highlighted in the abstract along with the depletion phenotype, was previously shown by Faubel et al. Moreover, if the authors want to make something of the apparent discrepancy with protein localization in Chlamydomonas or humans (Canning, Cell Rep 2022), they would need to demonstrate antibody specificity by staining their knockout MEFs.

5. CDKL5 and the CDKL family of kinases

On p5, the authors state "The Chlamydomonas genome encodes two proteins, CrCDKL5 and FAP262, that are similar to the CDKLs." While this may be true, their Fig 1A fails to make this point clearly. If this phylogenetic tree is to be taken at face value, Chlamydomonas encodes four other CDKL-related proteins, FLS1, FLS2, CrCRE09.g395658 and CrCe17.g709500, which are more similar to human CDKL1-4. This is inconsistent with the proposed evolutionary history of the CDKL protein family recently published by Martín-Carrascosa et al., Front Cell Dev Biol 2025 according to which chlorophytes possess only the ancestral CDKL5 protein (their Fig 2), a paper that incidentally would seem well worth citing. If these additional proteins are unrelated, a better tree ought to be shown. If not, and in particular given the similar function in ciliary length control reported for CDKL1 in C. elegans, their potential function ought to be examined.

Minor comments/text changes

p4

"Cilia and flagella (terms can be used interchangeably)"

I might be a little overly pedantic here, but while a flagellum is a type of cilium, not every cilium is a flagellum.

p8

"Similarly, in contrast to the partial rescue of swimming speed observed with wild-type CDKL5-GFP, neither of the mutated proteins rescued swimming speed (Fig 2C), indicating that CDKL5's kinase activity is necessary for normal motility. Consistent with the results for wild-type CDKL5-GFP, expression of the mutated proteins had little or no effect on beat frequency (Fig 2D)."

Contrary to what the authors imply, Fig 2D shows the two mutant rescues to have beat frequencies similar to the wild-type rescue, all of which are slightly improved over the lf5 mutant alone but none near that of controls.

p9

"Overexpression of the wild-type gene reduced the cilia length to slightly less than normal whereas expression of kinase-dead Cdkl5 did not alter cilia length but expression of Cdkl5 with mutated activation-loop phosphorylation sites was able to rescue. (Fig 3C)."

The statistical tests in Fig 3C should assess rescue ie compare to the mutant, not to wild-type. Further, the surprising finding that the activation loop mutant is able to rescue is not further remarked upon, even though just one paragraph later the authors comment on how important this phosphorylation is in humans (based on the literature) and Chlamydomonas (their findings, Fig 2)

p17

"Normal localization of mammalian CDKL5 is independent of its kinase activity but requires Cdk20"

This is somewhat counterintuitive. If CDK20 function is to activate CDKL5 kinase activity and it does so by phosphorylating the kinase on its activation loop, why does loss of CDK20 affect CDKL5 localization, but loss of CDKL5 kinase activity does not?

p18

"To determine if IFT of CDKL5 is dependent on its kinase activity, we similarly observed full-length flagella of lf5 cells expressing CDKL5K33R-GFP, CDKL5Y166F-GFP , or CDKL53A-GFP (Fig 9B, C, E)."

Somewhat misleading. 'Full-length' implies cilia to be morphologically normal, which in the mutant rescues they are not. "Mature" maybe?

p19

"The movement of CDKL5Y166F-GFP by IFT, together with CDKL5Y166F-GFP's concentration (albeit diffuse) at the flagellar base, suggests that CDKL5Y166F-GFP retains some kinase activity."

This seem a little bit of a stretch, to go from localization pattern to predictions of kinase activity.

p29

"Forty-two proteins were reduced in abundance in the absence of CDKL5. Some of these proteins have functions related to flagellar motility, providing a possible explanation for the reduced flagellar beat frequency observed in the lf5 mutant. Also of note, the length-regulatory kinase LF4 was reduced, suggesting that LF4's role in IFT is downstream of CDKL5."

While it is tempting to speculate that LF4 may be downstream of CDKL5, the mass spec data of the authors is not particularly conclusive evidence, especially considering Wang et al, FASEB 2019 showed LF4 localization to be normal in lf5 mutants.

Reviewer #2: This a well-written, compelling, tour de force on the regulation, localization, and function of a protein kinase that regulates ciliary/flagellar length in Chlamydomonas and vertebrates. Each of the many conclusions in the abstract about CDKL5 are solidly supported by one or more independent experimental strategies and approaches. After decades of research, the molecular mechanisms that regulate cilia/flagella length remain poorly understood. The findings in the manuscript make clear the challenges of investigating length control mechanisms: Length control depends on an exquisite interplay of multiple complex systems. These authors elegantly show that the length-regulating flagellar protein kinase, CDLK5, is phosphorylated at multiple sites through autophosphorylation and through the activity of at least two other protein kinases, and that alterations in CDLK5 phosphorylation influence its location, protein kinase activity, and function. Their comparisons between the vertebrate and Chlamydomonas proteins support common mechanisms of regulation, and the finding that a tubulin-transporting IFT protein is a substrate for CDLK5 provides a reasonable explanation for a central function of CDLK5 in length control.

This is a beautiful manuscript! I have 3 minor comments:

1) In the Figure 2 kinase activity experiments, it would be helpful to provide the results (shown later) that the mutant forms of the protein are expressed and present in flagella.

2) Figure 4 A should be made much larger. It shows important results that are well presented in the text but challenging to understand by viewing the figure.

3) The Discussion could be shortened. Although well written and easily understood, it sometimes devotes too much text to re-describing results.

Reviewer #3 (Tomoharu Kanie, identifies himself): CDKL5 is a protein kinase that belongs to cyclin dependent kinase family like family. CDKL5 mutations are found in Cdkl5 Deficiency Disorder (CDD), of which phenotypes show overlaps with the ones caused by cilia defects (i.e., ciliopathies). The molecular mechanisms underlying the defects caused by CDKL5 are not well understood. In this manuscript, the authors carried out detailed analyses of Chlamydomonas CDKL5, and its mammalian orthologue, and provide deep insights into the functions and regulations of CDKL5, which include the ciliary waveform regulations, the length control of mammalian primary cilia, comprehensive determination of phosphorylation sites within CDKL5, IFT modulations, and identification of potential substrates. The main texts are logically written, the data is of high quality, interpretation of data is mostly reasonable, statistical analyses were adequately performed, and legends and methods sections provide sufficient details. This paper will be of interest to biologists who work on cilia and flagella as well as geneticists who studies cilia-related disorders.

I think the manuscript is already in a good shape and should be published with only minor modifications. I have only a couple of minor comments for the data interpretation, and this should be solved by modification of the text rather than additional experiments.

Minor comments for the data

I was convinced by almost of all data except the ones that showed phosphorylation of CDKL5 activation loop is mediated by LF2. In Figure 5, the authors showed that peptides with S162/T164/Y166/S168 phosphorylation were almost undetectable in either wild-type or kinase-dead (K33R) CDKL5, while mono-phosphorylation at each of the above residues were found in CDKL5 purified from the alga lacking LF2. The authors hypothesized that CDKL5 may be highly phosphorylated at S162T164Y166 in wild-type cells and that the tri-phosphorylated form may not be detected by mass spec. To test the hypotheses, the authors performed mass spec analyses of CDKL5 before and after the phosphatase treatment, and found that the intensity of phosphopeptides as well as non-phosphopeptides were dramatically increased after CIP treatment, supporting the authors' hypothesis. Based on those results, the authors concluded that LF2 directly (or indirectly) phosphorylates the activation loop of CDKL5. I agree that the authors' hypothesis is one possibility, but there may be other explanations for the results. LF2 may phosphorylate CDKL5 at other positions than S162T164Y166, and the phosphorylation of the residue may inhibit the trypsin digest through the ionic bond with lysine/arginine. LF2 depletion or phosphatase treatment may block the phosphorylation and increase the efficiency of trypsin cleavage, which would result in increase in the intensity of the precursors. Besides, each of S162T164Y166 is highly phosphorylated in LF2 mutants, so it would not be easy to think that LF2 directly (or indirectly) phosphorylates the activation loop. I understand that figuring out what exactly is happening in those experiments is not trivial (the authors may possibly inject synthetic peptides that correspond to the trypsinized peptides that include activation loop with or without phosphorylation to prove the authors' hypothesis), and I do not think doing it is absolutely critical. Instead, please consider discussing alternative possibilities that could explain the results.

The data shown in Figure 6 is also complicated. The fact that CDKL5 3A (S162A, T164A, Y166A) mutant was not auto phosphorylated in vitro may suggest that the in vitro phosphorylation may occur at the S162/T164/Y166 residues. The failure of in vitro phosphorylation with the kinase-dead mutant (K33R) of CDKL5 confirmed that the in vitro phosphorylation in the CDKL5 was indeed mediated by CDKL5 itself (auto phosphorylation). These results may imply that the S162/T164/Y166 residues are the main sites of CDKL5 auto phosphorylation. However, this possibility conflicts with the data shown in figure 5, where the authors showed that the activation loop phosphorylation is likely NOT mediated by CDKL5 itself as there was no difference in the S162/T164/Y166 phosphorylation between wild-type and the kinase-dead mutant of CDKL5. Another possibility to explain the results is that the CDKL5 3A (S162A, T164A, Y166A) mutant lost its kinase activity. Perhaps, this is what the authors are thinking about. Figuring out what is exactly happening may require substantial amount of work, and that itself can be sufficient for a paper. Again, I do not think that is absolutely critical for this paper, but would like to ask the authors to explain their interpretations and discuss alternative possibilities in the main text. This would help readers to interpret the presented data correctly.

Other minor comments

(1) Page 5 "CDKL5 is a serine/threonine protein kinase": I was initially confused with the fact that CDKL5 may autophosphorylate at tyrosine residue within the activation loop, as the authors initially described CDKL5 as a serine/threonine protein kinase. Based on the fact, this kinase should be considered as a dual-specificity kinase. I think it would reduce confusion if the authors clearly state the word "dual specificity kinase' early on in the paper.

(2) Figure 3A: Figure legends indicated that the authors stain the cells with CEP164, but I did not clearly see CEP164 in the Figure. Also, asterisk and arrowhead were not explained in the figure legend.

(3) Figure 3B and C: I initially got confused with the label "MS234, MS238, GP1192", as those labels were not explained in the figure legend. Referring to Table S8 in the Figure legend would be helpful. Similarly, throughout the paper, the authors use unique labels for Chlamydomonas lines (e.g., g1, 21gr), MEF lines (e.g., 26548), and plasmids (e.g., MS234). I appreciate that the authors listed what exactly they used, and think that it would help readers to easily figure out what exactly those labels are if the authors refer them to the corresponding tables (e.g., Table1, table S8) in the corresponding figure legends.

(4) Figure 3B and 3C: It would be easier to understand what the 2A mutant is if the authors write the actual mutations (in this case, T183A and Y185A?)

(5) Figure 4A: I feel dots are too small, and "K" and "*" marks are difficult to find. Please use bigger fonts.

(6) Figure 4B: Maybe I am missing, but I feel I could not find the methods for the CIP treatment of the flagella samples (I was able to find the methods for the CIP treatment of the mass spec samples shown in Figure 5B).

(7) Figure 5Ba: I did not understand why S162T164, S162Y166, and Y166 are omitted in this table.

(8) Page 31 line 6-page 32: The discussion in these pages are highly overlapped with the discussion written in the prior pages. Please consolidate the discussion, so that readers can take the most important findings of the paper.

---

## [Decision Letter · Decision Letter 2]

27 Oct 2025

Dear Dr Pazour,

Thank you for your patience while we considered your revised manuscript entitled "LF2 phosphorylation of the CDKL5 activation loop controls CDKL5’s activity" for consideration as a Research Article at PLOS Biology. Your revised study has now been evaluated by the PLOS Biology editors, the Academic Editor and two of the original reviewers.

The reviews are attached below. You will see that the reviewers are mostly satisfied with the revision. However, Reviewer 1 thinks that you should add an analysis of ciliary lengths in CDK20/CDKL5 double mutants compared to the individual single mutants, given that these mutants have been generated during the revision. After consulting with the Academic Editor, we think that it would be important to address this point.

In light of the reviews, we are pleased to offer you the opportunity to address the remaining point raised by Reviewer 1 in a revision that we anticipate should not take you very long. We will then assess your revised manuscript and your response to the reviewers' comments with our Academic Editor aiming to avoid further rounds of peer-review, although we might need to consult with the reviewers, depending on the nature of the revisions.

**IMPORTANT - SUBMITTING YOUR REVISION**

3. Resubmission Checklist

a) *PLOS Data Policy*

b) *Published Peer Review*

Sincerely,

Ines

--

Ines Alvarez-Garcia, PhD

Senior Editor

PLOS Biology

Reviewers' comments

Rev. 1:

In my original review of the study on CDKL5 by Pazour and colleagues I criticized the lack of clarity of their manuscript (major point 1), which made it largely inaccessible to the reader. This was particularly unfortunate given the clear and pressing need to make the not insubstantial community of clinical researchers working on Cdkl5 Deficiency Disorder aware of the potential ciliary connection of their disorder. In preparing this revision, the authors have extensively reworked the text, which has improved considerably in my opinion. The one recommendation I would still have here given the aforementioned need to raise awareness is to incorporate cilia into the title and use vertebrate nomenclature, perhaps: "Phosphorylation by LF2/CDK20 controls activity of the ciliary kinase CDKL5".

The revised text now more clearly elaborates the manuscript's novel findings and also discusses CDKL5 in the context of the wider CDKL kinase family, which appears to share a related function. Nevertheless, given that this was supposed to have been a major revision, I would have expected at least some attempt to address my second major point concerning a lack of mechanistic insight with additional experiments. I understand the authors do not wish to expand the scope of their present study and investigate the relationship between CDKL5 and other CDKL kinases. However, especially considering the authors have actually generated CDK20/CDKL5 double mutants, an analysis of ciliary lengths in those mutants compared to the individual single mutants as strongly suggested in my original review was entirely within its present scope and would have considerably strengthened the manuscript without being particularly demanding. To me this is a striking omission that should still be corrected.

Concerning the other, minor, points raised in my original review, these have been adequately addressed.

Rev. 3: Tomoharu Kanie

The authors addressed all my comments. I appreciate it.

---

## [Editor Report · Decision Letter 3]

14 Nov 2025

Dear Dr Pazour,

Thank you for your patience while we considered your revised manuscript entitled "LF2/CDK20 phosphorylation of the activation loop regulates the ciliary kinase CDKL5" for publication as a Research Article at PLOS Biology. This revised version of your manuscript has been evaluated by the PLOS Biology editors and by the Academic Editor.

Based on our Academic Editor's assessment of your revision, we are likely to accept this manuscript for publication, provided you satisfactorily address the data and other policy-related requests stated below my signature.

In addition, we would like you to consider a suggestion to improve the title:

“The cyclin-dependent kinase CDK20 phosphorylates the activation loop of the ciliary kinase CDKL5 to control flagellar length”

We expect to receive your revised manuscript within two weeks.

*Published Peer Review History*

*Press*

Sincerely,

Ines

--

Ines Alvarez-Garcia, PhD

Senior Editor

PLOS Biology

Fig. 2A, C-E; Fig. 3C; Fig. 5A-D; Fig. 7D; Fig. 8B-D; Fig. 9D, F, I; Fig. 10B, D; Fig. 11B; Fig. 12B, D, E; Fig. S3B; Fig. S4C, D; Fig. S5B and Fig. S7

Please also ensure that figure legends in your manuscript include information on WHERE THE UNDERLYING DATA CAN BE FOUND, and ensure your supplemental data file/s has a legend. For example, you could add at the end of all the corresponding figure legends the following: "The data underlying the graphs shown in the figures can be found in S1 Data"

**Please also make the files you have deposited in the Pride database (PXD066796, PXD066877 and PXD068782) publicly available at this stage.

CODE POLICY

We require the original, uncropped and minimally adjusted images supporting all blot and gel results reported in an article's figures or Supporting Information files. We will require these files before a manuscript can be accepted so please prepare and upload them now. Please carefully read our guidelines for how to prepare and upload this data: https://journals.plos.org/plosbiology/s/figures#loc-blot-and-gel-reporting-requirements

While you might have added all in the file provided, it is in a format we cannot access. Please provide the raw gels for the following figures either in a word or a PDF file:

Fig. 4B, C; Fig. 6; Fig. 7A, B; Fig. S2A, B; Fig. S3A; Fig. S4A and Fig. S6A

---

## [Editor Report · Decision Letter 4]

26 Nov 2025

Dear Dr Pazour,

Thank you for the submission of your revised Research Article entitled "Activation of the ciliary kinase CDKL5 is mediated by the cyclin-dependent kinase CDK20/LF2 to control flagellar length" for publication in PLOS Biology. On behalf of my colleagues and the Academic Editor, Renata Basto, I am delighted to let you know that we can in principle accept your manuscript for publication, provided you address any remaining formatting and reporting issues. These will be detailed in an email you should receive within 2-3 business days from our colleagues in the journal operations team; no action is required from you until then. Please note that we will not be able to formally accept your manuscript and schedule it for publication until you have completed any requested changes.

PRESS

Sincerely, 

Ines

--

Ines Alvarez-Garcia, PhD

Senior Editor

PLOS Biology
